# KDM2B links the Polycomb Repressive Complex 1 (PRC1) to recognition of CpG islands

Anca M Farcas[1], Neil P Blackledge[1†], Ian Sudbery[2†], Hannah K Long[1,3†], Joanna F McGouran[4], Nathan R Rose[1], Sheena Lee[5], David Sims[2], Andrea Cerase[1], Thomas W Sheahan[1], Haruhiko Koseki[6], Neil Brockdorff[1], Chris P Ponting[2], Benedikt M Kessler[4], Robert J Klose[1*]

[1]Department of Biochemistry, University of Oxford, Oxford, UK; [2]CGAT, MRC Functional Genomics Unit, Department of Physiology, Anatomy and Genetics, University of Oxford, UK; [3]Molecular Haematology Unit, Weatherall Institute of Molecular Medicine, John Radcliffe Hospital, University of Oxford, Oxford, UK; [4]Ubiquitin Proteolysis Group, Central Proteomics Facility, Nuffield Department of Clinical Medicine, Centre for Cellular and Molecular Physiology, University of Oxford, UK; [5]Department of Physiology, Anatomy and Genetics, University of Oxford, Oxford, UK; [6]Laboratory for Developmental Genetics, RIKEN Research Center for Allergy and Immunology, Yokohama, Japan

**Abstract** CpG islands (CGIs) are associated with most mammalian gene promoters. A subset of CGIs act as polycomb response elements (PREs) and are recognized by the polycomb silencing systems to regulate expression of genes involved in early development. How CGIs function mechanistically as nucleation sites for polycomb repressive complexes remains unknown. Here we discover that KDM2B (FBXL10) specifically recognizes non-methylated DNA in CGIs and recruits the polycomb repressive complex 1 (PRC1). This contributes to histone H2A lysine 119 ubiquitylation (H2AK119ub1) and gene repression. Unexpectedly, we also find that CGIs are occupied by low levels of PRC1 throughout the genome, suggesting that the KDM2B-PRC1 complex may sample CGI-associated genes for susceptibility to polycomb-mediated silencing. These observations demonstrate an unexpected and direct link between recognition of CGIs by KDM2B and targeting of the polycomb repressive system. This provides the basis for a new model describing the functionality of CGIs as mammalian PREs.

*For correspondence:
rob.klose@bioch.ox.ac.uk

†These authors contributed equally to this work as second authors

## Introduction

The capacity to segregate and functionalize regulatory elements within large and complex vertebrate genomes relies on the activity of chromatin modifying and epigenetic systems (*Kouzarides, 2007*). Nevertheless, it remains very poorly understood how the enzymes responsible for initiating these functional chromatin states are targeted to and recognize defined regions of the genome. Understanding the molecular mechanisms that underpin these processes is becoming increasingly important given the mounting genetic evidence implicating misregulation of chromatin modifying activities in human disease and cancer (*Greer and Shi, 2012*; *Butler et al., 2012*). In mammals, the majority of CpG dinucleotides are methylated on the 5 position of cytosine and this epigenetic modification can be stably transmitted across cell divisions (*Klose and Bird, 2006*). DNA methylation is thought to be a highly stable modification that helps maintain intergenic and heterochromatic regions in a transcriptionally inert chromatin state (*Bird, 2002*; *Klose and Bird, 2006*). A major exception to

**eLife digest** Gene expression in eukaryotic cells can be controlled in a number of different ways, including various epigenetic mechanisms that do not involve making changes to DNA sequences that define the genes themselves. A well-known epigenetic mechanism for silencing genes in vertebrates is DNA methylation—the addition of a methyl group ($CH_3$) to cytosine, which is one of the four bases found in the DNA. Methylation is thought to silence genes by preventing transcription factors from binding to the DNA, and also by recruiting proteins that inhibit the transcription of DNA.

DNA methylation occurs naturally throughout the genome, mostly at positions where cytosine is bonded to guanine to form a CpG dinucleotide. While the cytosine bases in most CpG dinucleotides are methylated, there are short stretches of DNA known as CpG islands that contain a high proportion of unmethylated CpG dinucleotides. These islands contain a large number of cytosine and guanine bases, and they are often found at or near transcription start sites.

The lack of methylation at CpG islands has long been assumed to have a passive role in gene expression, leaving the DNA easily accessible and available for transcription factors to bind and initiate transcription. However, recent work suggests that CpG islands may have a more active role. In particular, it has been shown that specific proteins bind to CpG islands to create chromatin environments that are more favourable for the initiation of gene expression. Moreover, a subset of CpG islands can also bind polycomb-group proteins, including the polycomb repressive complex 1 (PRC1) that silence gene expression. These complexes have an important role in the regulation of genes during early development in animals, but the mechanism by which PRC1 recognizes CpG islands in mammals has remained enigmatic.

Farcas et al. now reveal that a protein, KDM2B (FBXL10), can recognize CpG islands and recruit PRC1 to them. To achieve this, KDM2B encodes a DNA binding domain that specifically recognizes non-methylated CpG dinucleotides. By interacting biochemically with a variant PRC1 complex, KDM2B then nucleates PRC1 at CpG islands, and PRC1 activity silences certain polycomb target genes in embryonic stem cells. Surprisingly, Farcas et al. also find low but appreciable levels of PRC1 at most CpG islands genome-wide, in addition to the high levels of PRC1 at selected islands: this suggests that KDM2B may sample the whole genome to find CpG islands where PRC1 can establish silencing. An improved understanding of the polycomb repressive system, and the role of CpG islands within it, could lead to new insights into the role of epigenetic mechanisms in mammalian development.

this pervasive methylation is short contiguous regions of the genome, called CpG islands (CGIs), which lack DNA methylation and are associated with roughly two-thirds of human gene promoters (*Illingworth and Bird, 2009*). It is thought that CGIs function to buffer gene regulatory elements from the repressive effects of surrounding DNA methylation and therefore play a central role in the regulatory capacity of the genome (*Blackledge and Klose, 2011*).

Although for the past two decades CGIs were considered to play a largely passive role in promoter and regulatory element function, we and others recently discovered that the non-methylated CpG dinucleotides found within CGIs act as a binding site for proteins that contain the ZF-CxxC DNA binding domain (*Blackledge et al., 2010*; *Thomson et al., 2010*). ZF-CxxC domain-containing proteins are found in chromatin modifying complexes and therefore play an unexpected and proactive role in specifying unique chromatin modification architecture at CGI associated gene promoters. For example, the ZF-CxxC domain containing protein CFP1 is a core component of the SET1 histone H3 lysine 4 (H3K4) methyltransferase complex (*Lee and Skalnik, 2005*) and targets this transcriptionally permissive modification to CGIs genome-wide. Similarly, the KDM2A histone H3K36 demethylase enzyme contains a ZF-CxxC domain that targets the enzyme directly to CGIs where it removes histone H3 lysine 36 (H3K36) dimethylation (*Blackledge et al., 2010*). In mammals H3K36me1/me2 is found broadly across the genome (*Peters et al., 2003*; *Robin et al., 2007*; *Blackledge et al., 2010*) and may be inhibitory to transcriptional initiation (*Strahl et al., 2002*; *Carrozza et al., 2005*). Specific targeting of KDM2A to CGIs appears to remove H3K36me2 at CGIs as part of a mechanism to mark these regions with transcriptionally permissive chromatin (*Blackledge et al., 2010*). Importantly, the

targeting of these modifications to CGI associated gene promoters generally occurs independently of the transcriptional state of the associated gene (*Blackledge et al., 2010*). This suggests that ZF-CxxC mediated chromatin modifications function upstream of transcription to create a permissive chromatin environment at gene promoters and demarcate these regions from surrounding non-regulatory regions of the genome (*Blackledge and Klose, 2011*; *Deaton and Bird, 2011*).

Unlike the majority of CGIs which are marked with a permissive chromatin architecture controlled by the ZF-CxxC dependent chromatin modification system, a subset of CGIs can exist in an alternative more repressed chromatin state. This is dictated by the action of the two central polycomb repressive complexes (PRCs) in mammals, called PRC1 and PRC2 (*Mikkelsen et al., 2007*; *Simon and Kingston, 2009*). PRC1 complexes have E3 ubiquitin ligase activity that mono-ubiquitylates H2A on position 119 (H2AK119ub1) (*Wang et al., 2004b*; *de Napoles et al., 2004*) and PRC2 complexes have histone H3 lysine 27 (H3K27) methyltransferase activity (*Cao et al., 2002*; *Czermin et al., 2002*; *Kuzmichev et al., 2002*; *Müller et al., 2002*). In contrast to the permissive and accessible chromatin state found at most CGIs, polycomb mediated chromatin modifications are thought to generate a more compact and inhibitory chromatin environment, limiting transcription of associated genes (*Shao et al., 1999*; *Lavigne et al., 2004*; *Francis et al., 2004*; *Eskeland et al., 2010*).

In *Drosophila*, which lack CGIs, polycomb repressive complexes are targeted to regions of the genome called polycomb response elements (PREs) through the action of transcription factors (reviewed in (*Schuettengruber et al., 2007*; *Ringrose and Paro, 2007*; *Simon and Kingston, 2009*)). Interestingly, in mammals this situation is very different in that PREs predominantly correspond to CGIs (*Ku et al., 2008*; *Woo et al., 2010*) and there is mounting evidence that chromatin features specific to CGIs may play an important role in guiding PRC complexes to these particular regions of the genome (*Mendenhall et al., 2010*; *Kanhere et al., 2010*; *Lynch et al., 2012*). At the mechanistic level, how CGIs can function as mammalian PREs remains unknown, constituting a major gap in our understanding of how CGIs are functionally linked to the polycomb repressive systems in mammals.

In addressing this question, here we demonstrate that KDM2B, a paralogue of KDM2A, plays an important role in targeting the polycomb repressive complex 1 (PRC1) to CGIs and regulating gene expression in mouse embryonic stem cells (ESCs). This provides the first evidence linking direct recognition of CGIs with recruitment of a polycomb repressive complex in mammals.

## Results

### KDM2B binds specifically to non-methylated CpG dinucleotides and associates with CGIs genome-wide

We recently demonstrated that the KDM2A protein binds to CpG islands genome-wide (*Blackledge et al., 2010*). However, its paralogue, KDM2B, was reported to be concentrated in the nucleolus. Therefore, to examine the properties of endogenous KDM2B and understand how it functions in comparison to KDM2A, an antibody against KDM2B was raised. In contrast to previous reports, we observe that KDM2B is distributed throughout the nucleus and excluded from nucleoli (*Figure 1—figure supplement 1*). Based on these observations, and considering that KDM2B also encodes a ZF-CxxC domain, we set out to examine whether KDM2B had a similar role to KDM2A in binding non-methylated DNA and CGIs (*Blackledge et al., 2010*). In agreement with previous work, recombinant KDM2B was found to bind non-methylated DNA in vitro (*Koyama-Nasu et al., 2007*; *Yamagishi et al., 2008*; *Blackledge et al., 2012*) in a manner that is comparable to that of KDM2A (*Figure 1A,B*). This suggests that KDM2B, like KDM2A, may recognize non-methylated DNA in vivo and bind CGI elements. To examine this possibility a chromatin immunoprecipitation sequencing (ChIP-seq) analysis was carried out to determine the binding profiles of KDM2A and KDM2B genome-wide in mouse ESCs. A visual inspection of these profiles suggests that both KDM2A and KDM2B localize to promoter associated CGIs and are excluded from non-CGI associated promoters (*Figure 1C*). When KDM2A and KDM2B ChIP-seq signal was plotted over all transcription start sites (TSSs) that had been segregated based on their classification as CGI or non-CGI associated there was a very specific enrichment of both KDM2A and KDM2B at the CGI associated class of transcription start site (*Figure 1D*). Furthermore, when KDM2A and KDM2B signal was plotted over all CGIs, including those away from gene promoters, there was a very specific spatial localization of both proteins at CGIs with signal intensity peaking at the centre of the island (*Figure 1E*). Together these observations demonstrate in vivo that KDM2A and B are both bound to CGIs genome-wide.

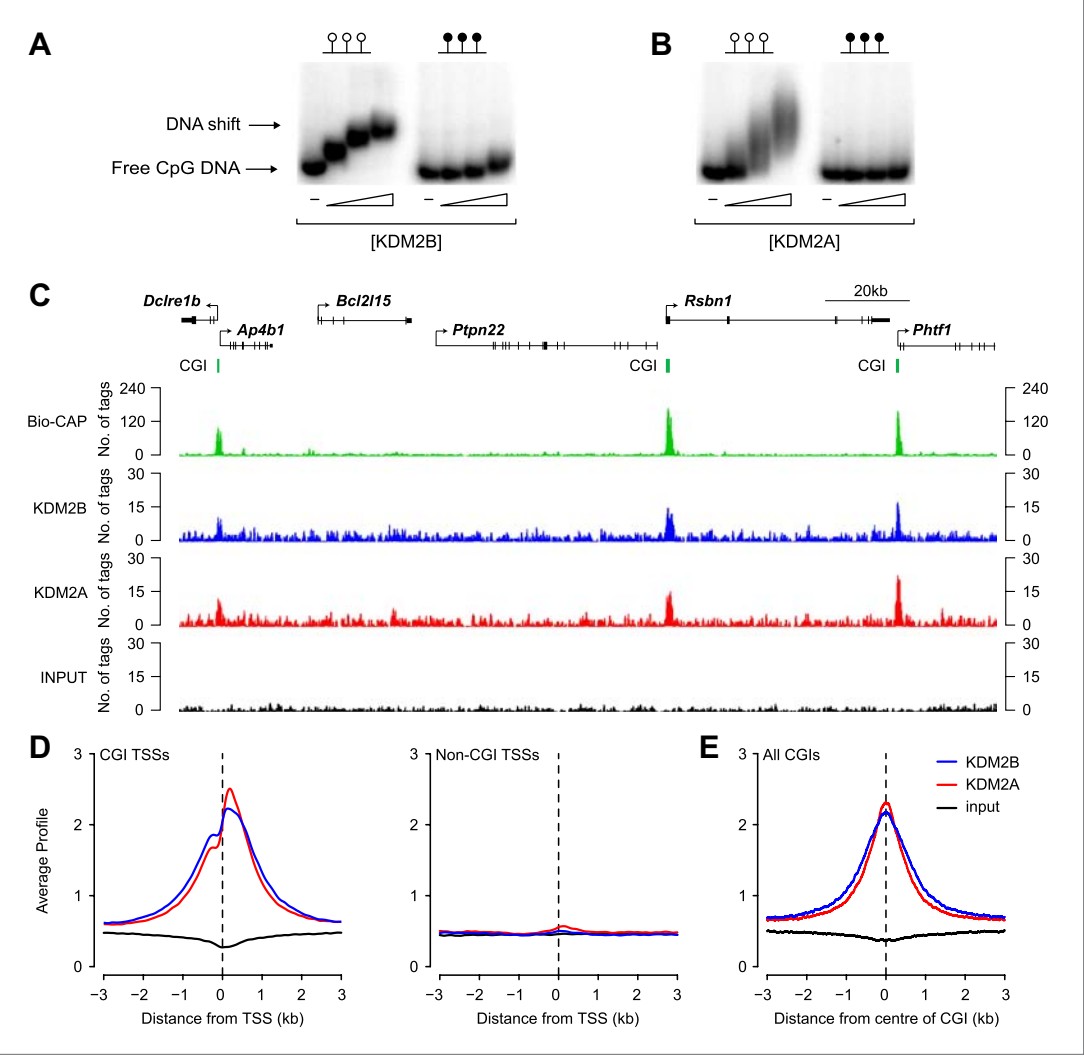

**Figure 1**. KDM2B binds to non-methylated CpG DNA and localises to CGIs genome-wide. (**A**) Electrophoretic mobility shift assay (EMSA) demonstrating that the recombinant KDM2B ZF-CxxC domain binds to a DNA probe containing non-methylated CpGs (left panel) in a concentration-dependent manner. Binding is abrogated by CpG methylation (right panel). (**B**) An analogous EMSA to part (**A**) using the known CGI binding factor KDM2A for comparison. (**C**) Input, KDM2A, and KDM2B ChIP-seq profiles over a region of the genome containing CGI and non-CGI associated genes. Bio-CAP-seq profiles are shown to indicate the location of non-methylated DNA (***Blackledge et al., 2012***). Above the sequencing traces individual genes are shown with the arrow indicating the transcription start site and vertical black lines corresponding to exons. The location of CGIs are indicated by green bars. Both KDM2A and KDM2B associate specifically with CGIs containing non-methylated DNA. (**D**) KDM2A and KDM2B ChIP-seq signal segregates specifically with CGI associated gene promoters (left panel) and is excluded from gene promoters not associated with CGIs (right panel). (**E**) KDM2A and KDM2B ChIP-seq signal is centred over CGIs, in agreement with their capacity to recognize non-methylated DNA at these sites.
The following figure supplements are available for figure 1.

**Figure supplement 1**. KDM2B is found throughout the nucleus and not concentrated in the nucleolus.

## KDM2B is enriched and KDM2A depleted at polycomb occupied CGIs

KDM2A and KDM2B profiles at CGI associated genes appeared largely similar (***Figure 1***). However when the levels of KDM2A and B were directly compared at individual CGIs genome-wide, a subset of CGIs was found that were preferentially enriched for KDM2B and depleted of KDM2A (***Figure 2A***). To begin examining whether there is a specific feature shared amongst KDM2B enriched CGIs, the genes

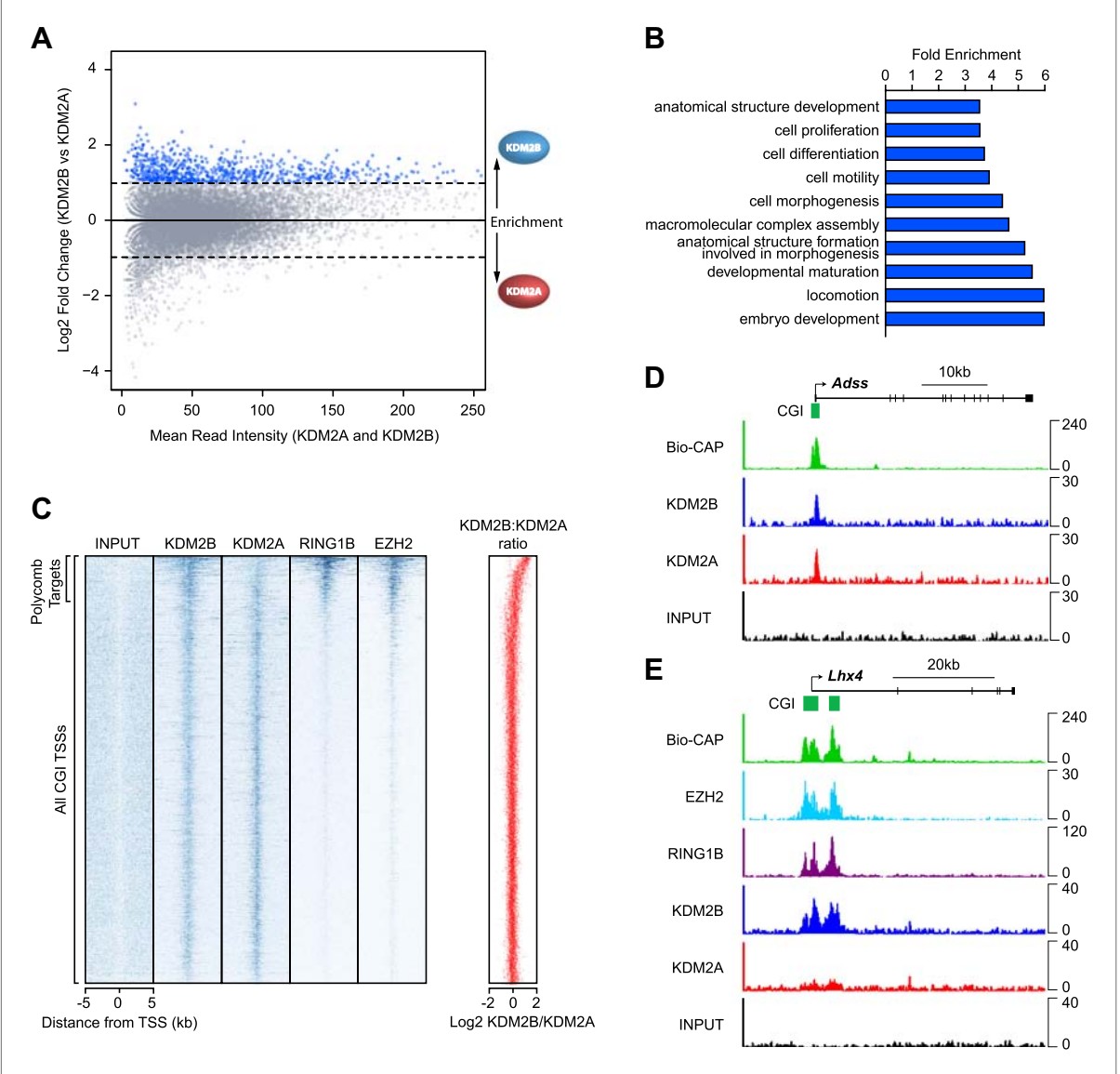

**Figure 2**. KDM2B is enriched at polycomb repressed CGIs. (**A**) An MA-like plot depicting the relative enrichment of KDM2A and KDM2B at all gene associated CGIs in mouse ESC cells. The log2 mean read intensity is displayed on x-axis and the log2 relative enrichment of KDM2B compared to KDM2A is displayed on the y-axis. The subset of CGIs highly enriched for KDM2B is coloured blue. (**B**) A histogram displaying fold enrichment values for GO term analysis of the genes which are over twofold enriched for KDM2B (blue data points, part (**A**)) at a FDR <5%. (**C**) A heat map illustrating ChIP-seq signal at all CGI transcription start sites (TSSs) in mouse ESC cells indicating depletion of KDM2A and enrichment of KDM2B at polycomb target CGIs marked by RING1B and EZH2. The scatter plot (far right in red) illustrates the log ratio of KDM2B to KDM2A enrichment at the same intervals depicted in the heat map as a scatter plot. (**D**) ChIP-seq profiles for KDM2A and KDM2B at the *Adss* non-polycomb target CGI gene indicating similar binding of both KDM2A and B (upper panels). (**E**) ChIP-seq for KDM2A, KDM2B, RING1B (PRC1), and EZH2 (PRC2) at the *Lhx4* gene (lower panels). KDM2B is specifically enriched and KDM2A depleted at this polycomb repressed CGI. In all cases Bio-CAP-seq indicates the location of underlying non-methylated DNA and clearly depicts the spatial relationship between KDM2B, polycomb group proteins, and non-methylated DNA.

The following figure supplements are available for figure 2.

**Figure supplement 1**. KDM2B is enriched at polycomb associated CGIs.

associated with these CGIs were extracted and subjected to gene ontology (GO) enrichment analysis (*Figure 2B*). Interestingly, these genes were significantly associated with GO terms related to embryo development, morphogenesis, and cellular differentiation. In early development and embryonic stem cells, these classes of genes are often targeted by the polycomb repressive complexes and maintained

in a poised but silent state until their expression is induced during lineage commitment (*Lee et al., 2006*; *Boyer et al., 2006*; *Bracken et al., 2006*; *Mikkelsen et al., 2007*). The enrichment of KDM2B at these specific classes of genes suggests that KDM2B may play a unique role in their regulation perhaps related to polycomb mediated repression.

To examine this possibility, the binding profiles of KDM2A, KDM2B, and components of the polycomb repressive complexes were examined in more detail. In mammals, there are two general polycomb repressive complexes (PRC) called PRC1 and PRC2 (*Shao et al., 1999*; *Cao et al., 2002*; *Czermin et al., 2002*; *Kuzmichev et al., 2002*; *Müller et al., 2002*; *Levine et al., 2002*). Previously, ChIP-seq based analysis has been used extensively to examine the location of PRC1 (via RING1B) (*Tavares et al., 2012*) and PRC2 (via EZH2) genome-wide (*Peng et al., 2009*). Using this information, KDM2A and KDM2B ChIP-seq was compared to the polycomb repressive complexes in mouse ESCs by examining the signal intensity around all CGI associated gene promoters (*Figure 2C*). From the heat maps it was immediately apparent that KDM2A and KDM2B bind in a seemingly equal manner to most CGI associated genes, but a subset of genes are specifically enriched for KDM2B and depleted of KDM2A. In keeping with the gene ontology analysis, the KDM2B enriched target genes segregate almost exclusively with genes enriched for polycomb repressive complex members RING1B and EZH2. Perhaps more surprisingly, the profile of the PRC1/2 components over CGI-associated target genes did not map precisely to the gene TSS but instead tracked with KDM2B binding and the underlying non-methylated DNA signal at these same regions. This striking spatial relationship is apparent at individual genes and also more generally when PRC1/2 components are examined at all polycomb bound genes (*Figure 2C–E* and *Figure 2—figure supplement 1*). Together these observations suggest that KDM2B, unlike KDM2A and CFP1 (*Thomson et al., 2010*), is enriched at polycomb associated CGIs. Furthermore, the clear spatial relationship between polycomb repressive proteins, KDM2B, and non-methylated DNA indicates there may be a mechanistic relationship between recognition of non-methylated DNA at CGIs and polycomb repressive complex nucleation.

## KDM2B forms a variant PRC1 complex characterized by the PCGF1 subunit

Based on the clear enrichment of KDM2B at polycomb associated CGIs in ESCs in vivo, we sought to understand if KDM2B is part of a protein complex in ESCs that might contribute to this localization. To achieve this we isolated stable ESC lines expressing epitope-tagged KDM2B and carried out affinity purification from nuclear extract followed by mass spectrometry to identify associated proteins (*Figure 3A–C*). To ensure that any identified interactions were not mediated through DNA or non-specific interactions with the affinity matrix, we also carried out parallel purifications in which the extract had been pre-treated with nuclease to remove any DNA contamination and from a cell line containing only the empty expression vector. In the purifications from extracts containing epitope-tagged KDM2B a series of proteins in addition to KDM2B were identified (*Figure 3B,C*). These included RING1B, YAF2, RYBP, PCGF1, BCOR, and BCORL1. Importantly RING1B is the E3 ubiquitin ligase that functions as the catalytic core of the PRC1 complex and is responsible for mono-ubiquitylation of H2A at position 119 (H2AK119ub1). To verify that endogenous KDM2B interacts with RING1B, KDM2B was immunoprecipitated from ESC nuclear extract and the presence of RING1B in the immunoprecipitates verified by western blotting (*Figure 3D*). Therefore, KDM2B exists as part of a variant PRC1 complex in mouse embryonic stem cells. This observation is in agreement with recent work in cancer cell lines that identified KDM2B as part of a similar variant PRC1 complex (*Gearhart et al., 2006*; *Sánchez et al., 2007*; *Gao et al., 2012*).

To understand in more detail the composition and specificity of the KDM2B/PRC1 complex in ESCs, we generated individual stable cell lines expressing epitope tagged RING1B, YAF2, RYBP, and PCGF1 (*Figure 3E* and *Figure 3—figure supplement 1*). Nuclear extracts were prepared from each cell line, epitope-tagged factors were affinity purified, and the associated proteins identified by mass spectrometry. Importantly KDM2B was identified in all purifications validating the association between KDM2B and these PRC1 components. Recently it has become apparent that different PRC1 complexes form in a manner that is dependent on a central dimerization partner that interacts with the RING1A/B catalytic core (*Gao et al., 2012*). These dimerization partners, called polycomb group ring fingers (PCGFs), act as bridging molecules that convey unique subunit composition to RING1B containing complexes and are required for H2AK119ub1 catalysis (*Cao et al., 2005*; *Li et al., 2006*; *Ben-Saadon et al., 2006*; *Buchwald et al., 2006*) (*Figure 3F*). There are six characterized RING1B dimerization

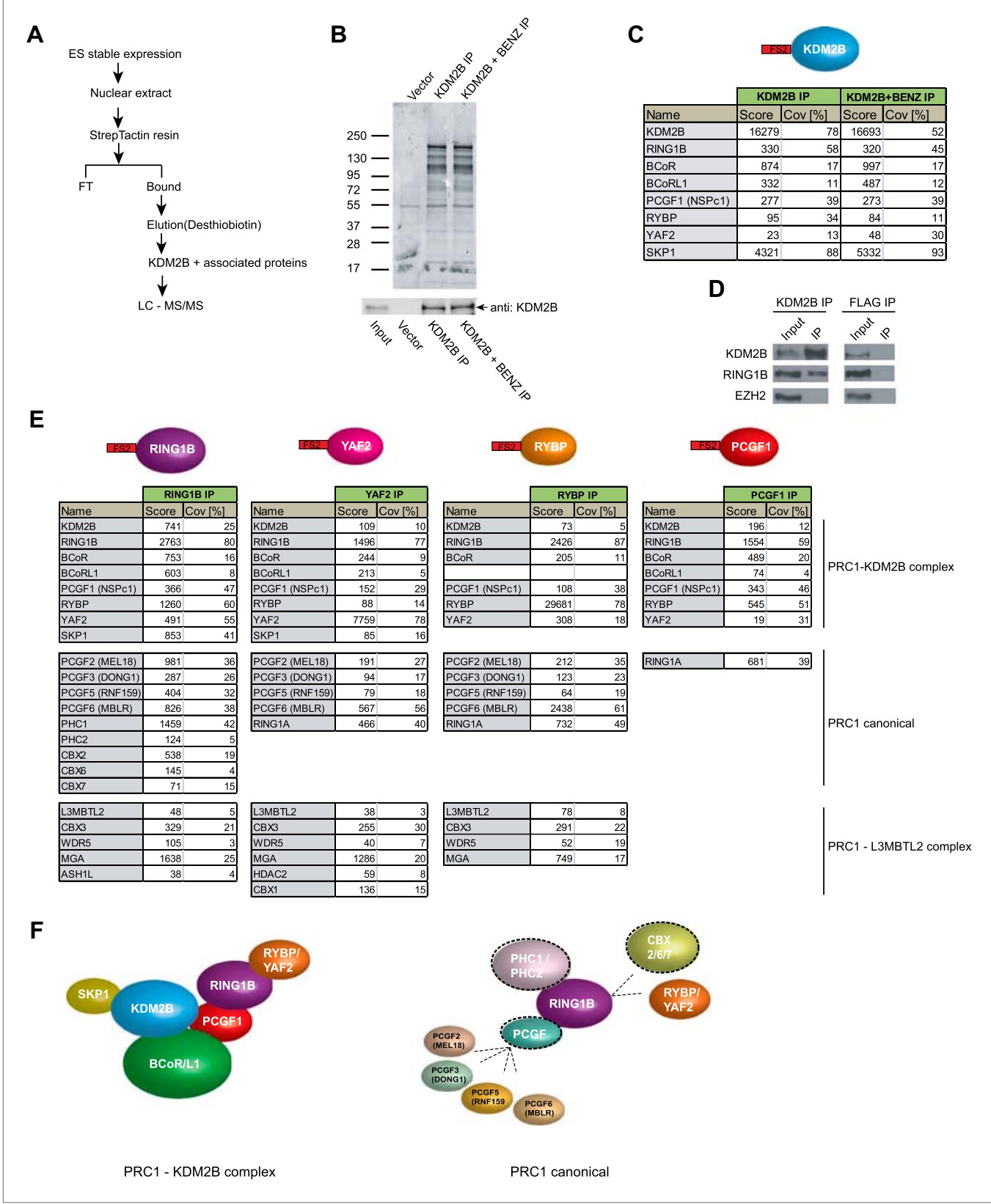

**Figure 3**. KDM2B forms a variant PRC1 complex in mouse ESCs containing RING1B and PCGF1. (**A**) To purify KDM2B and associated proteins, a mouse ESC cell line stably expressing Flag-2XStrepII-tagged KDM2B was generated. Nuclear extract was isolated from this cell line, KDM2B affinity purified, *Figure 3. Continued on next page*

*Figure 3. Continued*
and the purified proteins subject to mass spectrometry. (**B**) Purified KDM2B fractions were resolved by gradient SDS-PAGE and visualized by SyproRuby staining. The purifications were performed in the absence and presence of benzonase to exclude DNA-mediated interactions and a cell line containing only the empty vector was used to control for non-specific binding to the affinity matrix. The elutions were probed by western blot for KDM2B as indicated. (**C**) Elutions from the KDM2B affinity purification were directly analysed by tryptic digestion followed by peptide identification by LC–MS/MS. The Mascot scores and peptide coverage are shown for the respective affinity purifications. KDM2B in ESCs associates with a variant PRC1 complex containing RING1B, BCOR/BCORL1, PCGF1, RYBP, YAF2 and SKP1. (**D**) Western blot analysis of endogenous KDM2B immunoprecipitation from ESC nuclear extract, verifying that KDM2B interacts with RING1B, but not the PRC2 component EZH2. Flag immunoprecipitation was performed as negative control. (**E**) Reciprocal affinity purifications and subsequent LC-MS/MS for RING1B, YAF2, RYBP, and PCGF1 confirm the interaction between KDM2B and these PRC1 components. This analysis further indicates that PCGF1 is unique to the KDM2B-PRC1 complex. Protein identification scores and sequence coverage (Cov [%]) are indicated. (**F**) A schematic representation of the variant KDM2B PRC1 complex (left panel) in comparison to canonical PRC1 complexes (right panel).
The following figure supplements are available for figure 3.
**Figure supplement 1**. KDM2B forms a variant PRC1 complex in mouse ESCs containing RING1B and PCGF1.

partners PCGF1 (NSPC1), PCGF2 (MEL18), PCGF3 (DONG1), PCGF4 (BMI1), PCGF5 (RNF159), and PCGF6 (MBLR). In purifying RING1B complexes we identified five of the six known PCGF subunits and most of the major auxiliary factors known to interact with these complexes (*Figure 3E*). This supports previous observations that RING1B is the central enzymatic component of most PRC1 complexes in ESCs (*Illingworth et al., 2012*). Although YAF2 and RYBP were both identified in our KDM2B purification, when these proteins were purified independently it became apparent that they are also found in multiple other PCGF containing PRC1 complexes (*Figure 3E*). This indicates that both YAF2 and RYBP can form part of the KDM2B-PRC1 variant complex, but also function as part of other PRC1 complexes (*Figure 3E*). YAF2 and RYBP are thought to interact directly with RING1B (*Wang et al., 2010*) suggesting that they are likely to be recruited to the KDM2B-PRC1 complex via RING1B. In the KDM2B purification the only PCGF identified was PCGF1, suggesting that this factor is unique to the KDM2B-PRC1 variant complex. When the PCGF1 complex was purified (*Figure 3E*) its composition was almost identical to the KDM2B complex, with the exception that it also included RING1A, an alternative catalytic core of PRC1 that is highly sequence-similar to RING1B but is expressed at greatly reduced levels in ESCs. Therefore, based on our detailed reciprocal complex purifications, KDM2B forms a PRC1 variant complex that includes BCOR, BCORL1, PCGF1, RING1, and YAF2 or RYBP (*Figure 3F*). Interestingly, in agreement with PRC1 purifications from cancer cell lines (*Gao et al., 2012*), the PCGF1-containing complex associates with RYBP or YAF2 but fails to integrate chromobox domain-containing (CBX) proteins that recognize H3K27me3 (*Cao et al., 2002*; *Wang et al., 2004a*). CBX proteins and RYBP/YAF2 are thought to interact with RING1B in a mutually exclusive manner (*Wang et al., 2010*). This suggests that PCGF1 may play an important role in specifying the capacity of RYBP and YAF2 to associate with RING1B in the KDM2B variant PRC1 complex. Importantly, the inclusion of KDM2B in a PRC1 complex in ESCs suggest that the relationship between KDM2B and polycomb repressed CGIs may be mediated through its direct interaction with PRC1 components.

## KDM2B relies on its ZF-CxxC domain and not its interaction with PRC1 to bind polycomb repressed CGIs

Both KDM2A (*Figure 2C*) and CFP1 (*Thomson et al., 2010*) are inhibited from binding to polycomb repressed CGIs in vivo despite the presence of non-methylated CpG at these sites. This may be related to the compact nature of polycomb-repressed regions and the distinct requirement for the ZF-CxxC domain to bind accessible linker DNA (*Zhou et al., 2012*). However, KDM2B differs from KDM2A and CFP1 in that it is enriched at polycomb repressed CGIs (*Figure 2*). Based on our biochemical purifications that indicate KDM2B in ESCs is part of a variant PRC1 complex (*Figure 3*), it remained possible that KDM2B was enriched at polycomb repressed CGIs via a ZF-CxxC domain independent mechanism relying on PRC1. To test this hypothesis we obtained mouse ESCs which lack RING1A and have a floxed allele of RING1B that can be conditionally removed by addition of the drug tamoxifen (*Figure 4A*) (*Endoh et al., 2008*). Removal of the catalytic core of PRC1 is known to destabilize the remainder of the core PRC1 complex (*Leeb and Wutz, 2007*; *Endoh et al., 2008*). Therefore, we first isolated chromatin from wild type and RING1A/B deleted ESC cells and carried out chromatin immunoprecipitation for RING1B and KDM2B. As would be expected, there were

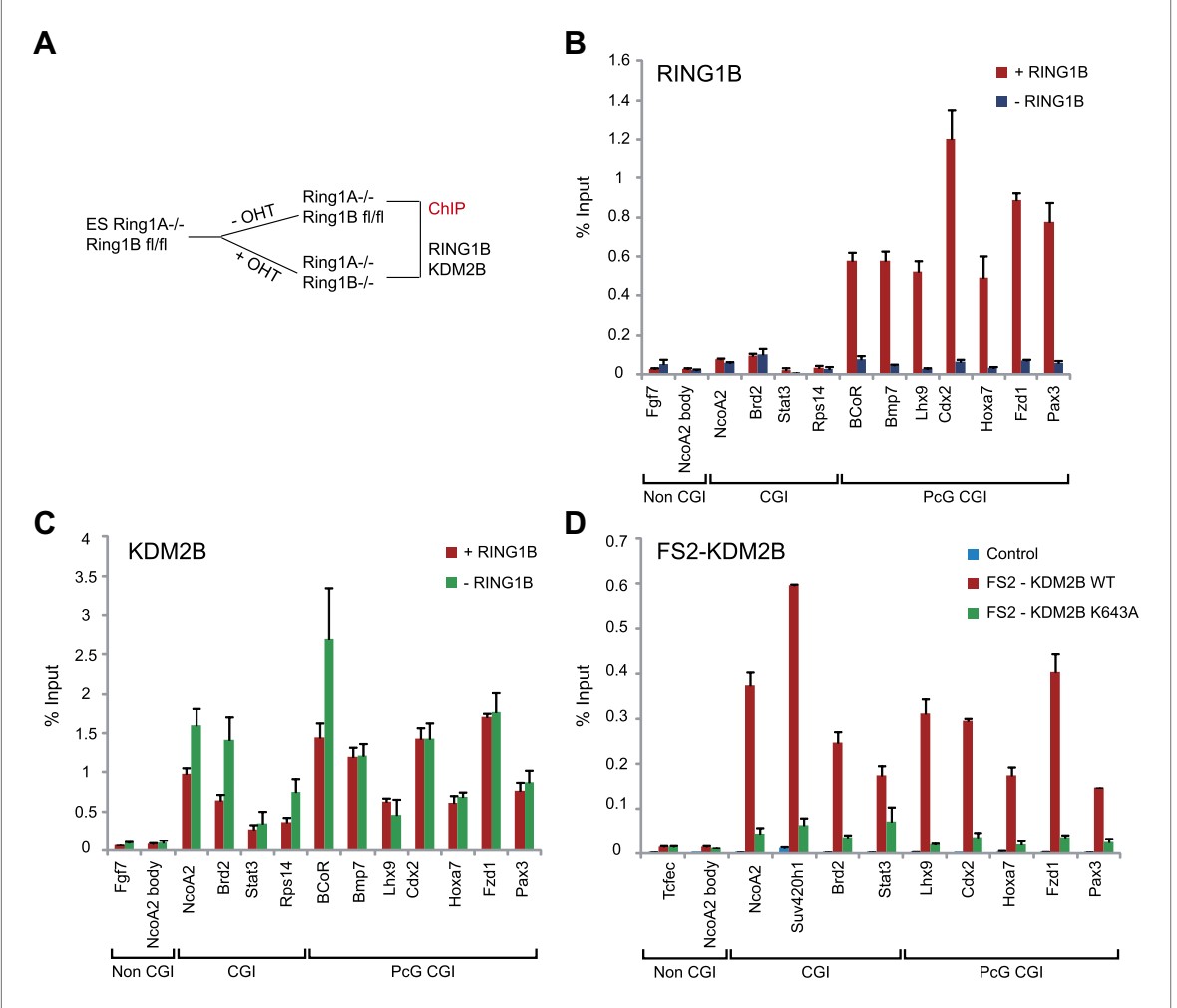

**Figure 4**. KDM2B ZF-CxxC DNA binding domain is required for CGI binding. (**A**) Schematic representation showing removal of RING1B in ESC Ring1a⁻/⁻ Ring1bᶠˡ/ᶠˡ cells by tamoxifen treatment. (**B**) ChIP analysis indicating that RING1B is enriched at polycomb target CGIs in untreated cells (red bars). After 48 hr of tamoxifen treatment RING1B binding is lost at polycomb targets (blue bars). Error bars represent SEM of three biological replicates. (**C**) ChIP analysis demonstrating that removal of RING1B does not lead to loss of KDM2B binding at regular or polycomb associated target CGIs (compare red and green bars). This demonstrates that RING1B is not required to recruit KDM2B to polycomb repressed sites. Error bars represent SEM of three biological replicates. (**D**) ChIP analysis in ESC cells stably expressing tagged wild-type (WT) KDM2B or a mutant KDM2B that disrupts its DNA-binding capacity (K643A). The ZF-CxxC domain of KDM2B is required for KDM2B binding to CGIs regardless of their polycomb status. Error bars represent SEM of two biological replicates.

high levels of RING1B at polycomb repressed CGI containing genes and the RING1B signal was lost following tamoxifen treatment (*Figure 4B*). We then analysed the levels of KDM2B at a series of polycomb repressed and non-polycomb repressed CGI associated genes by ChIP and observed only small or no changes in KDM2B binding in the absence of RING1A/B (*Figure 4C*). This suggests that an intact PRC1 complex is not required to target KDM2B to polycomb repressed sites in vivo.

Given that KDM2B does not appear to rely on an intact PRC1 complex to localize to polycomb CGIs, we hypothesized its recognition of polycomb targets must be achieved at least in part through the function of its ZF-CxxC non-methylated DNA binding domain. To test this hypothesis we generated stable ESC lines expressing either epitope tagged wild type KDM2B or a version of KDM2B containing a point mutation in the ZF-CxxC domain that ablates binding to CpG dinucleotides. We then isolated chromatin from these cell lines and analysed binding of exogenous KDM2B to CGI targets (*Figure 4D*). Importantly, WT KDM2B localized specifically to CGIs but a ZF-CxxC mutant failed to bind any class of CGI. This demonstrates that although KDM2B is part of a PRC1 complex

its localisation to regular and polycomb repressed CGIs requires the recognition of non-methylated DNA via the ZF-CxxC domain. The capacity of KDM2B to target CGIs in the absence of PRC1 components suggests that KDM2B could function as targeting module for PRC1 and may contribute to polycomb mediated repression.

## Depletion of KDM2B leads to up-regulation of a subset of polycomb repressed target genes

Based on the clear enrichment of KDM2B at polycomb repressed genes and the inclusion of KDM2B in a PRC1 complex we examined whether depletion of KDM2B would affect the expression of polycomb repressed gene targets in mouse ESCs. To achieve this we stably knocked down KDM2B in ESCs using a lentiviral mediated shRNA approach. Knockdown was approximately by 80% at the RNA level (*Figure 5A*) and there was a clear reduction of KDM2B at the protein level (*Figure 5B*). Importantly, depletion of KMD2B did not result in destabilization of either the PRC1 or PRC2 complexes allowing us to specifically examine defects resulting from reduced KDM2B (*Figure 5B*). Microarray based gene expression profiling was then carried out to compare gene expression in the scrambled and KDM2B shRNA lines (*Figure 5C*). Gene expression profiling revealed a substantial number of expression changes with genes being both down-regulated and up-regulated. In total 653 genes changed expression by more than 1.5-fold: 329 had reduced gene expression and 324 had increased gene expression. Based on the association of KDM2B with polycomb repressed genes we then focussed in on the set of 324 genes that are up-regulated in the KDM2B depletion line. Interestingly 24% of these genes are characterized by RING1B occupancy. This is significantly greater than would be expected by chance (1.9-fold enrichment, hypergeometric p=$1.3 \times 10^{-8}$) and is similar to the percentage of up-regulated genes in the RING1B null ESC cells that are PRC1 targets (*Leeb et al., 2010*). Furthermore, when all gene expression changes are depicted in a volcano plot there is a clear propensity for RING1B occupied genes to have increased gene expression when KDM2B is depleted (*Figure 5C*). In order to validate the observed up-regulation of polycomb targets in the KDM2B depletion cell lines we analysed a panel of 12 genes determined to be up-regulated based on the microarray using quantitative RT-PCR (*Figure 5D*). In this analysis we observed corresponding increases in transcription in the KDM2B depletion line at these polycomb targets. Together these findings indicate that as with the removal of RING1B, depletion of KDM2B leads to up regulation of a subset of polycomb repressed target genes in mouse embryonic stem cells.

## KDM2B targets RING1B to CGIs

The up-regulation of a subset of polycomb repressed genes following KDM2B depletion and its physical association with a PRC1 complex suggests that KDM2B may play a role in polycomb function at CGIs, perhaps acting as a targeting factor for the PRC1 complex via its capacity to recognize CGI DNA. To examine this possibility we carried out ChIP-seq analysis for RING1B and KDM2B in the control and KDM2B knockdown cell lines. As expected KDM2B occupancy was greatly reduced in the knockdown cell line, in fitting with the global reduction of KDM2B at the RNA and protein level (*Figure 6A*). Next we examined RING1B occupancy in the control cell line. We confirmed high levels of RING1B at previously identified RING1B sites (*Figure 6B* e.g. *Atf3* gene). Interestingly however, based on our highly optimized RING1B ChIP protocol we also observed a large number of lower magnitude binding events ('novel' sites) that did not correspond to sites previously characterized as being occupied by RING1B ('established' sites) (*Figure 6B*). To verify that these are bona fide RING1B binding events we carried out conventional ChIP followed by quantitative PCR on a series of these novel peaks in cells where we can conditionally delete RING1B and observed a clear loss of RING1B signal following deletion (*Figure 6—figure supplement 1*). This indicates these lower magnitude peaks correspond to novel RING1B binding events. Interestingly, 68% of the 15,740 newly identified RING1B binding sites overlapped with CGIs, and 77% of these were coupled to transcription start sites. Importantly, a minority of these novel RING1B occupied CGI sites showed detectable EZH2 (23%) or H3K27me3 (20%), indicating that these sites are largely devoid of appreciable PRC2. Including the already characterized RING1B associated CGIs, this means that clear RING1B peaks are detectable at nearly 70% of the 22,849 mouse CGIs genome-wide (*Illingworth et al., 2010*), and 88% of RING1B occupied CGIs also are bound by detectable levels of KDM2B.

To examine whether KDM2B is required for RING1B association with CGIs, RING1B ChIP-seq signal from the control and KDM2B KD cell lines was initially plotted over previously identified RING1B CGI

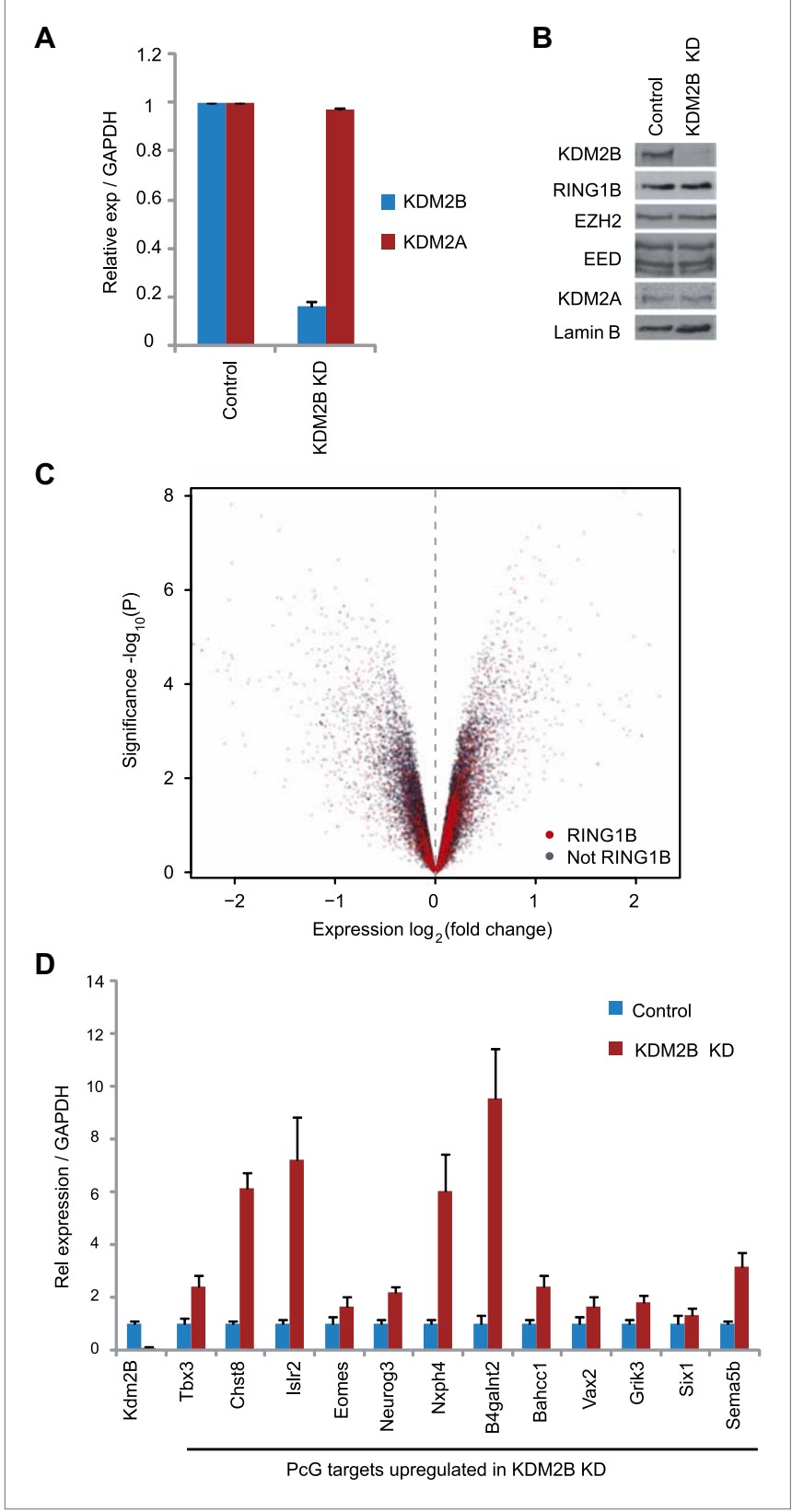

**Figure 5**. KDM2B depletion in ESCs results in the up-regulation of a subset of polycomb repressed target genes. (**A**) RT-PCR showing reduction of KDM2B at the mRNA level in the knockdown (KD) but not control cell line. The *Figure 5. Continued on next page*

*Figure 5. Continued*

levels of the closely related KDM2A mRNA were not affected. Error bars represent SEM of three biological replicates. (**B**) Western blot analysis showing that KDM2B knockdown depletes KDM2B but not the closely related KDM2A protein. Importantly depletion of KDM2B does not destabilize other polycomb group proteins including RING1B, EZH2 or EED. Lamin A/C indicates equal loading. (**C**) A volcano plot illustrating the gene expression changes in the KDM2B depletion line compared to the control cell line. The x-axis corresponds to fold change and the y-axis to the significance level. RING1B associated genes are coloured red and show preferential enrichment in the genes up-regulated upon KDM2B knockdown. (**D**) A panel of polycomb target genes identified in microarray analysis that were up-regulated upon depletion of KDM2B were validated by RT-PCR analysis. In most cases the level of up-regulation was more pronounced when analysed by RT-PCR. Error bars represent SEM of four biological replicates. Values are normalized to *Gapdh* and expression in the control line set to 1.

associated target sites (*Figure 6C*). This revealed a specific reduction in RING1B signal at these sites following KDM2B knockdown in fitting with a role for KDM2B in RING1B targeting. This reduction in RING1B binding was evident when the ChIP-seq signal was visualized at individual target genes (*Figure 6E*) and similar changes in RING1B occupancy were observed by conventional ChIP q-PCR analysis at selected RING1B bound CGIs (*Figure 6—figure supplement 2*). Importantly, at sites where RING1B occupancy was affected, loss of binding correlated with the underlying non-methylated DNA signal and KDM2B occupancy (*Figure 6E*). These observations indicate that depletion of KDM2B causes a loss of normal RING1B targeting at polycomb occupied CGI regions of the genome, in accordance with the observed reactivation of a subset of polycomb repressed genes in the KDM2B knockdown line.

Based on the observation that RING1B is more widely associated with CGI elements than previously appreciated (*Figure 6B*), we next examined whether KDM2B contributes to RING1B binding at these novel sites by plotting RING1B signal in the control and KDM2B knockdown cell lines over these regions (*Figure 6D,F*). Interestingly, depletion of KDM2B resulted in an even more dramatic effect on RING1B occupancy at these novel low magnitude sites, suggesting that these RING1B binding events are more dependent on KDM2B. Importantly, the observed reduction in RING1B occupancy at high and low magnitude sites was not due to destabilization of polycomb repressive systems, as the protein levels of both PRC1 and PRC2 components were similar in the wild type and KD cell lines (*Figure 5B*). Together these observations indicate that KDM2B plays a widespread role in targeting RING1B in vivo both at previously identified established polycomb-repressed CGI-associated genes and a novel class of lower magnitude RING1B occupied CGI sites.

## KDM2B is required for normal H2AK119ub1 levels

A major function of the PRC1 complexes is to mono-ubiquitylate histone H2A on position 119 (H2AK119ub1) and this activity is essential for PRC1 mediated gene repression (*Endoh et al., 2012*). A monoclonal antibody for H2AK119ub1 (*Vassilev et al., 1995*) has been widely used to evaluate H2AK119ub1 in vivo. Unfortunately, this antibody performs poorly under some experimental conditions, notably chromatin immunoprecipitation. To overcome this limitation we generated a novel antibody against a branched peptide encompassing the isopeptide junction between H2AK119 and the Ub chain. Following immunization, a double affinity purification strategy yielded a highly specific H2AK119ub1 antibody that recognizes native H2AK119ub and functions in ChIP (*Figure 7—figure supplements 1–3*). Using this novel H2AK119ub1 specific antibody we set out to examine whether depletion of KDM2B affected PRC1 catalysed chromatin modification in vivo. To achieve this we first compared the global level of H2AK119ub1 in the WT and KD cell lines (*Figure 7A,B*). Interestingly, depletion of KDM2B results in an approximately 40% global reduction of this modification, indicating that KDM2B contributes significantly to H2AK119ub1 at the genome scale, most likely due to its widespread role in guiding RING1B occupancy. This was not the result of more general changes in chromatin modification as other histone marks including H3K4me3 and H3K27me3 were largely unaffected (*Figure 7C*).

To understand whether these global reductions in H2AK119ub1 were also evident at polycomb repressed target genes, we analysed H2AK119ub1 at a panel of CGI associated genes. Interestingly reductions in H2AK119ub1 appeared to be restricted to the polycomb target CGIs that were up-regulated in the absence of KMD2B (*Figure 7D*). This also correlated with a more subtle reduction in H3K27me3

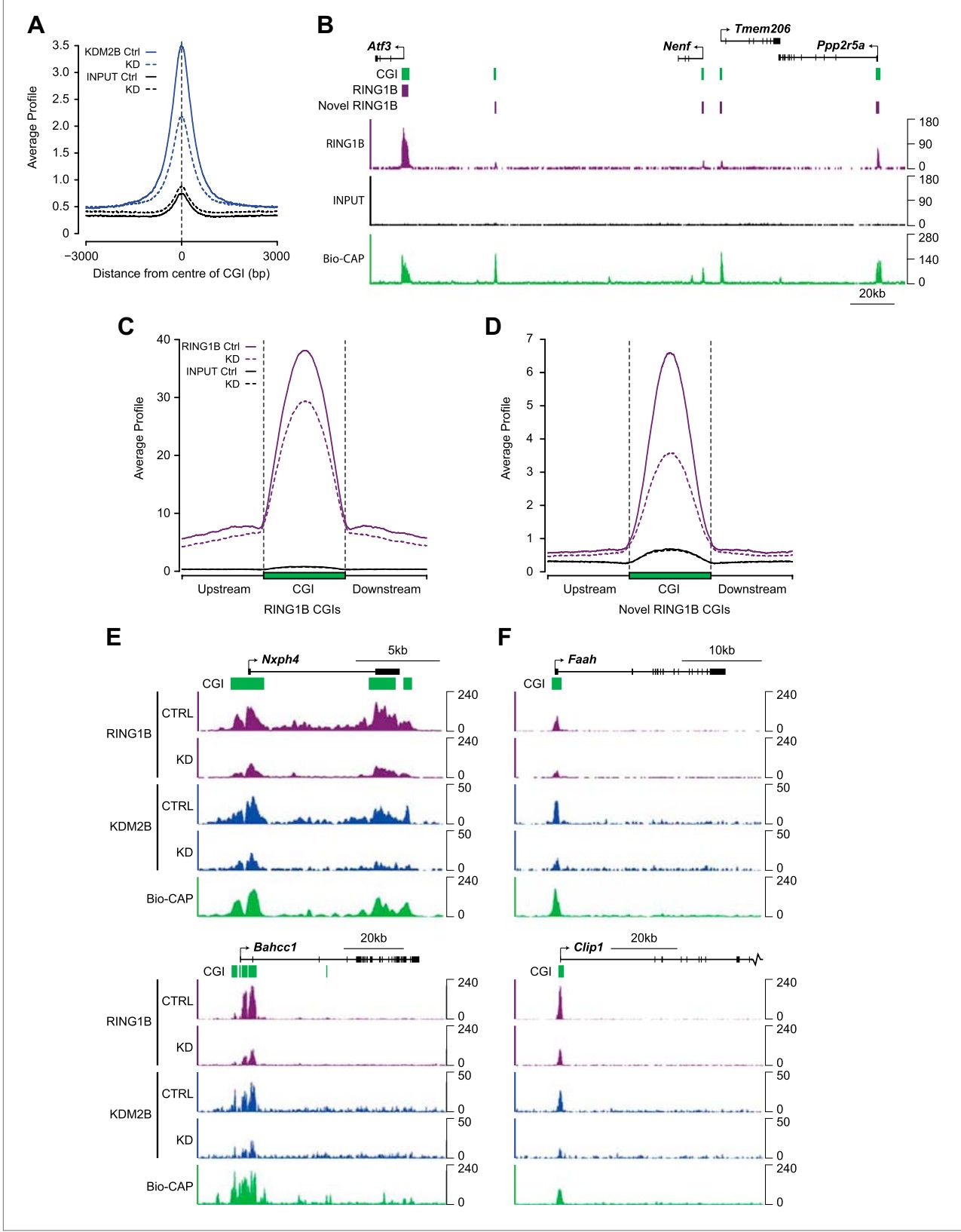

**Figure 6**. Depletion of KDM2B causes a reduction in RING1B occupancy. (**A**) KDM2B ChIP-seq signal was plotted at CGIs in the control (solid blue line) and knockdown cell line (dotted blue line). Sequencing signal in the input samples over the same regions are indicated as solid or dotted black lines. *Figure 6. Continued on next page*

*Figure 6. Continued*

KDM2B ChIP-seq signal is specifically lost over CGIs genome-wide in the KDM2B knockdown cell line. (**B**) A snapshot showing RING1B ChIP-seq signal in the control cell line illustrating a previously identified high magnitude RING1B binding sites (i.e. *Atf3*) and novel low magnitude RING1B binding sites. Input sequencing traces over the same region are shown in black. Bio-CAP-seq signal indicates regions containing non-methylated DNA. Above the sequencing traces gene promoters are show by black arrows and exons by vertical black lines. CGIs are shown as green bars with previously identified RING1B peaks and novel RING1B peaks indicated with purple boxes. (**C**) RING1B ChIP-seq signal from the control (solid purple line) and KDM2B knockdown cell line (dotted purple line) were plotted over previously identified CGI associated RING1B binding sites. Input sequencing signal was plotted over same regions. In the KDM2B knockdown line there is a specific reduction of RING1B binding over the CGI. (**D**) The same ChIP-seq signal as in part (**C**) was plotted over novel low magnitude binding sites identified in part (**B**). In the KDM2B knockdown cell line there is an even more severe loss of RING1B binding at these novel RING1B occupied sites. (**E**),(**F**) Examples of ChIP-seq profiles at individual high (**E**) and low (**F**) magnitude RING1B binding sites showing clear reduction of RING1B following KDM2B knockdown.

The following figure supplements are available for figure 6.

**Figure supplement 1**. Conditional removal of RING1B validates the identity of novel peaks identified in the RING1B ChIP-seq.

**Figure supplement 2**. KDM2B depletion results in a loss of RING1B at polycomb associated CGIs.

**Figure supplement 3**. Genes associated with novel RING1B CGIs are on average expressed at higher levels than genes associated with established RING1B CGIs.

but largely unchanged levels of H3K4me3 (*Figure 7D*). This suggests that although KDM2B plays a role in the nucleation of RING1B at most polycomb associated CGIs, the level of RING1B reduction at individual genes likely dictates the effects on H2AK119ub1 and gene expression. This is in accordance with the recent observation that H2AK119ub1 is important for gene repression mediated by PRC1 (*Endoh et al., 2012*). Interestingly, the level of H2AK119ub1 reduction at individual loci did not appear as dramatic as the global reductions observed by western blotting. One possibility for this difference may be that pervasive KDM2B dependent targeting of PRC1 to CGIs leads to low level deposition of H2AK119ub1 away from established polycomb sites. One can envisage how loss of KDM2B would more dramatically affect this additional pool of H2AK119ub1 resulting in significant global reduction in its abundance. Together these observations suggest that some polycomb repressed sites rely more specifically on KDM2B mediated RING1B occupancy for normal H2AK119ub1 and polycomb mediated gene repression (*Endoh et al., 2012*).

## Discussion

Here we demonstrate that KDM2B recognizes CGI elements via its ZF-CxxC domain and in contrast to its paralogue, KDM2A, is preferentially enriched at polycomb-repressed CGIs in mouse ESCs (*Figure 2*). By virtue of its physical association with a variant PRC1 complex (*Figure 3*) and through the functionality of its ZF-CxxC domain (*Figure 4*), KDM2B is required for normal targeting of the catalytic component of the PRC1 complex, RING1B, to CGI elements (*Figure 6*). Depletion of KDM2B results in a reduction of RING1B at polycomb repressed CGIs and leads to diminished global H2AK119ub1 (*Figure 7*). This causes reactivation of a subset of polycomb repressed target genes (*Figure 5*). Surprisingly, we also uncover a more widespread association of RING1B with CGIs that is also dependent on KDM2B (*Figure 6*). Together these observations provide an unexpected and direct link between CGIs, recognition of non-methylated DNA by KDM2B, and binding of the PRC1 complex to target sites in vivo.

The mechanisms underpinning how polycomb repressive complexes are recruited to target sites in mammals have remained largely elusive, with the exception that CGIs appear to function as the mammalian equivalent of the *Drosophila* polycomb response element (*Simon and Kingston, 2009*). In most species PRC1 is thought to be mechanistically recruited in a hierarchical manner through chromo-domain mediated recognition of H3K27 methylation at pre-existing PRC2 modified sites (*Cao et al., 2002*; *Min et al., 2003*; *Wang et al., 2004b*). This largely places PRC1 as a subservient reader of silencing events initiated by PRC2. However, the singularity of the hierarchal recruitment mechanism has recently been challenged by a series of elegant experiments in mammals demonstrating that although genetic perturbation of the PRC2 complexes reduces PRC1 localization to polycomb

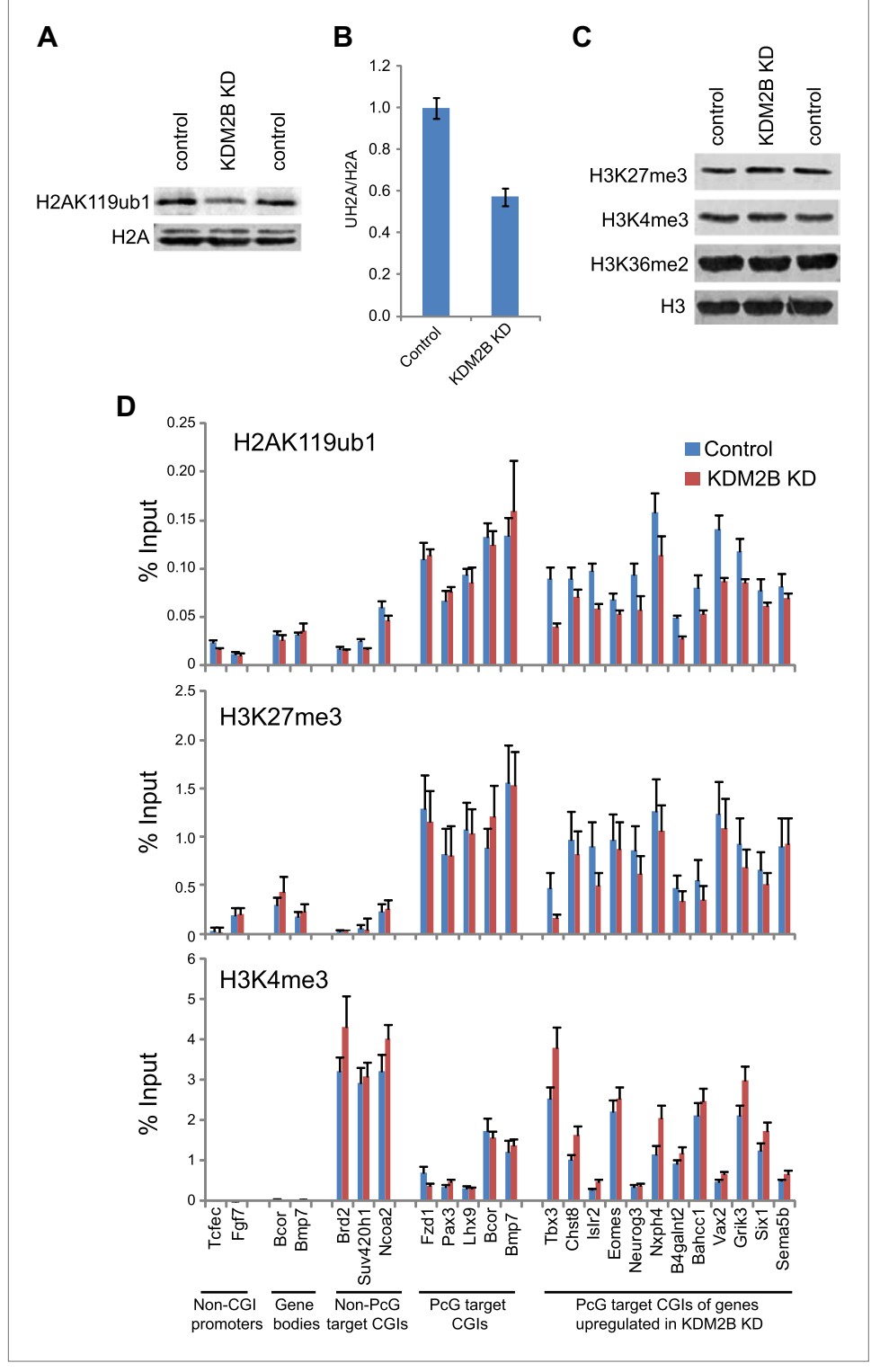

**Figure 7**. Depletion of KDM2B causes a reduction in H2AK119ub1. (**A**) KDM2B knockdown results in a global loss of H2AK119ub1, as demonstrated by western blot for H2AK119ub1 in KDM2B knockdown and control cells. Western blot for total H2A is shown as a loading control. (**B**) Quantification of H2AK119ub1 levels by fluorescence based quantitative western blotting. The levels of H2AK119ub1 are approximately 40% lower in KDM2B knockdown compared to control cells. The western blot signal for H2AK119ub1 was quantified relative to H2A and the error bars represent the SEM of six biological replicates. (**C**) KDM2B knockdown does not cause global changes in levels

*Figure 7. Continued on next page*

*Figure 7. Continued*

of H3K27me3, H3K4me3 or H3K36me2, as demonstrated by western blot in the KDM2B knockdown and control cells. Western for total Histone H3 is shown as a loading control. (**D**) KDM2B knockdown cells show locus specific depletion of H2AK119ub1 at genes up-regulated in the KDM2B knockdown cell line. Comparatively there are only small changes in H3K27me3 and H3K4me3 at these same sites. ChIP material was analysed by qPCR using primers specific for (i) non-CGI promoters, (ii) gene bodies, (iii) Non-PcG target CGIs, (iv) PcG target CGIs, and (v) PcG target CGIs of genes up-regulated in KDM2B knockdown cells. Error bars represent the SEM of four biological replicates.

The following figure supplements are available for figure 7.

**Figure supplement 1**. Purification of an H2AK119ub1 antibody.

**Figure supplement 2**. The purified H2AK119ub1 antibody recognizes a Xist inactivated chromosome.

**Figure supplement 3**. The purified H2AK119ub1 antibody works in chromatin immunoprecipitation.

repressed CGIs (*Tavares et al., 2012*), H3K27 independent PRC1 targeting mechanisms are sufficient to maintain H2AK119ub1 and repression at many PRC1 bound targets (*Schoeftner et al., 2006*; *Pasini et al., 2007*; *Leeb et al., 2010*; *Tavares et al., 2012*). The specific molecular mechanisms underpinning H3K27me3 independent PRC1 targeting nevertheless remain unclear. For example, a PRC1 variant complex containing RYBP and lacking chromobox domain-containing proteins appears to be involved in this process but it is unclear how RYBP would achieve specificity for polycomb targets as it has no inherent chromatin or DNA binding specificity (*Arrigoni et al., 2006*; *Neira et al., 2009*; *Tavares et al., 2012*; *Gao et al., 2012*; *Hisada et al., 2012*). Furthermore RYBP depletion appears to destabilize the RING1B protein (*Tavares et al., 2012*), reducing the effective pool of the enzyme and making it difficult to determine the specific contribution of RYBP to targeting.

Based on our observations that KDM2B is important for RING1B binding to CGI elements, the KDM2B-PRC1 complex provides an alternative mechanism for PRC1 recognition of polycomb repressed CGIs in mammals. KDM2B forms a variant PRC1 complex lacking CBX proteins but interestingly contains RYBP or the related YAF2 protein. In contrast to depletion of RYBP, reduction of KDM2B did not destabilize RING1B, allowing us to demonstrate that KDM2B contributes to the localization of RING1B to target sites in vivo. This targeting event appears to play an important role in the repression of a subset of polycomb target genes through H2AK119 ubiquitylation. Together these observations provide the first mechanistic evidence for the existence of a PRC1 targeting pathway that does not rely on recognition of H3K27me3 but instead interrogates the underlying non-methylated DNA sequence present at CGIs. Based on this work we propose an alternative to the hierarchical model of PRC1 targeting whereby CGIs are recognized by a variant PRC1 complex containing KDM2B through direct binding to non-methylated DNA. Other direct recognition mechanisms based on interactions between site specific DNA binding transcription factors and PRC1 have recently been identified further supporting the concept that DNA information at polycomb repressed sites is important for PRC1 targeting in vivo (*Ren and Kerppola, 2011*; *Dietrich et al., 2012*; *Yu et al., 2012*). Together, the hierarchical recruitment system and alternative direct targeting mechanisms likely function in a complementary and in some cases redundant fashion at many polycomb targets in vivo. This is supported by the observation that KDM2B depletion in embryonic stem cells results in up-regulation of only a subset of PRC1 target genes. Therefore, one can envision that polycomb repression, which must be maintained over cell divisions and developmental time scales, may rely on the combined activity of these separate pathways as an important stability component for maintenance of the silenced state.

KDM2B is enriched at polycomb repressed sites in vivo, but genome-wide analysis revealed it is also more broadly associated with CGIs. Based on this observation, an interesting question remains regarding how KDM2B specifically contributes to PRC1 recognition of polycomb targets, while also binding to CGIs throughout the genome. Our surprising observation that RING1B is detectable at nearly 70% of all CGIs is supportive of a model whereby KDM2B may act as a sampling module allowing PRC1 to be recruited at very low levels to most or all CGIs by virtue of its capacity to recognize CGIs. This would then imply that productive PRC1 enrichment and polycomb mediated silencing is in turn

only achieved at a subset of these sampled targets. Fittingly, novel RING1B associated CGIs do not differ in expression from the average gene (*Figure 6—figure supplement 3*), whereas established polycomb CGIs show reduced expression. This is largely in agreement with recent mechanistic work in mammalian systems studying the capacity of CGIs to attract polycomb mediated repression, where it has been suggested that the default state for CGIs is indeed to acquire polycomb mediated silencing but local chromatin features and other activating marks appear to function to counteract this process (*Mendenhall et al., 2010*; *Lynch et al., 2012*). Conceptually this parallels the long held view from genetic and molecular studies in *Drosophila* that indicate a continual antagonism exists between the function of positively acting trithorax chromatin modifying systems and repressive polycomb systems at PREs (*Poux et al., 2002*; *Klymenko and Müller, 2004*). In *Drosophila*, ultimately this balance is resolved by the local activity of transcriptional regulators. In analogy to the situation at *Drosophila* PREs, the KDM2B-PRC1 complex may therefore function to continually recruit PRC1 to CGIs at low levels but ultimately only establish the fully PRC1/PRC2 occupied and polycomb repressed state if the appropriate local chromatin and transcriptional environment is met. It is tempting to speculate that PRC1, and perhaps even PRC2 via its poorly characterized DNA binding components (*Cao et al., 2002*; *Kim et al., 2009*; *Li et al., 2010*), may be continually interfacing CGIs in a dynamic manner in order to constantly sample their susceptibility to polycomb repression. Once the conditions are met for polycomb mediated silencing, one can envision a simple feed forward loop may exist where the combined activities of PRC1 and PRC2 stabilize each other through canonical PRC1 complexes that contain CBX proteins recognising H3K27me3 placed by PRC2 (*Cao et al., 2002*; *Wang et al., 2004b*) and PRC2 activity favouring the increasingly compacted chromatin state associated with PRC1 occupancy (*Francis et al., 2004*; *Eskeland et al., 2010*; *Yuan et al., 2012*). This could account for the remarkable spatial correlation between mammalian polycomb repression domains and CGIs at established polycomb repressed sites (*Ku et al., 2008*). This speculative model is appealing as it would in part explain the co-occupancy of PRC1 and PRC2 at most established polycomb sites in vivo and could reconcile why only certain CGIs are susceptible to acquiring full polycomb complex nucleation and silencing in vivo.

In order to fully realize the complexities and interplay between polycomb silencing activities, more detailed molecular and biochemical understanding of this system is clearly required. For example, our study has focussed on ESCs where polycomb factors are highly abundant, so it will be imperative to understand if similar KDM2B dependent PRC1 targeting mechanisms function in more committed cell types. Also, it is known that H3K27me3 and CBX dependent PRC1 targeting activities play an important role in RING1B occupancy. Therefore, KDM2B dependent targeting of the PCGF1-PRC1 complex on its own is insufficient to account for the functionality of PRC1 in vivo. This suggests that a complex set of PRC1 targeting mechanisms, perhaps utilizing specific PRC1 complexes, and based on both direct and indirect recognition of CGI associated sites is required for normal PRC1 function. In agreement with this idea, knockdown of KDM2B leads to reactivation of only a subset of polycomb repressed genes. Although this may be due to incomplete removal of KDM2B by RNAi, it may also be related to compensatory function of other PRC1 complexes. Nevertheless, our demonstration that KDM2B links PRC1 to recognition of CGIs provides an important new piece of the puzzle and demonstrates for the first time a direct mechanism for polycomb recognition of CGIs elements in vivo. Importantly, it also provides a novel conceptual framework on which to consider initiation of polycomb mediated silencing.

## Materials and methods

### DNA constructs
The full length human KDM2B, mouse RING1B, mouse YAF2 (IMAGE clone 3488691), human PCGF1 (IMAGE clone 3621400), mouse RYBP (IMAGE clone 8861345) were PCR-amplified and inserted via ligation independent cloning (LIC) into a pCAG-IRES-puro eukaryotic expression vector that has been modified to express a N-terminal Flag and 2XStrep2 tag and contain LIC cloning sites. All PCR generated constructs were verified by sequencing. Mutation of the KDM2B ZF-CxxC DNA binding domain (K643A) was introduced into the wild type KDM2B construct by Quikchange mutagenesis XL kit (Stratagene, Santa Clara, CA).

### Antibodies
A polyclonal antibody against KDM2B was generated by immunizing a rabbit (PTU/BS Scottish National Blood Transfusion Service) with a Hisx6-tag fusion protein encoding amino acids 755–917 of human

KDM2B protein. KDM2B antigen was coupled to AffiGel10 resin (BioRad, Hemel Hempstead, UK) and the antibody was affinity purified. The RING1B mouse monoclonal antibody has been described previously (*Atsuta et al., 2001*). To obtain a highly pure RING1B antibody preparation, hybridoma cells were cultured and the antibody was purified from tissue culture supernatant by protein A agarose (RepliGen, Waltham, MA) based affinity chromatography. A rabbit polyclonal antibody against Flag-2XStrepII sequence was generated by immunization of a rabbit with a synthetically synthesized peptide conjugated to mariculture KLH carrier protein (ThermoScientific, Waltham, MA). Flag-2XStrepII peptide was covalently immobilized on SulfoLink resin (ThermoScientific) and antibody was affinity purified prior to use. Polyclonal antibody against histone H3 and histone H3 lysine 4 trimethylation (H3K4me3) were generated using the synthetic peptides CIQLARRIRGERA, and ART (K)QTARKSTGGC (where brackets indicate position of trimethyl mark), respectively. For both antibodies, peptides were conjugated to mariculture KLH carrier protein (ThermoScientific) prior to rabbit immunization, and after obtaining serum from immunized animals the antibodies were affinity purified using the respective peptide antigen coupled to SulfoLink resin (ThermoScientific). A polyclonal antibody against histone H3 lysine 36 dimethylation (H3K36me2) has been described previously (*Blackledge et al., 2010*), and commercially available antibodies were used to detect Histone H3K27 trimethylation (H3K27me3) (Diagenode, Liege, Belgium; 069-050) and total histone H2A (Millipore, Billerica, MA; 07-146).

## H2AK119ub antibody production

A branched peptide was synthesised (by GL Biochem, Shanghai, China) with the sequence $H_2N$-CVLLPKK ($H_2N$-LRGG)TESHHK-$NH_2$, where the $H_2N$-LRGG branch is coupled to the Lys119 residue through a Lys–Gly isopeptide bond. This branched peptide was conjugated to maleimide-activated KLH (Pierce Protein Biology Products, Rockford, IL) and a rabbit was immunised with the peptide-conjugate KLH.

Following immunisation H2AK119ub1 specific antibodies were purified in a two-step process. First, a peptide of the sequence $H_2N$-CVLLPKKTESHHK-$NH_2$, corresponding to amino acids 114–125 of histone H2A, was immobilised on SulfoLink Resin (Pierce Protein Biology Products). Serum was purified over this resin to remove antibodies cross-reactive with unmodified H2A. Secondly, the flow-through from this resin was then applied to a second Sulfolink Resin on which was immobilised the original branched peptide ($H_2N$-CVLLPKK [$H_2N$-LRGG]TESHHK-$NH_2$) used for immunisation.

## Cell culture

Murine V6.5 ESC cells (C57BL/6 (F) × 129/sv (M)) were cultured on Mitomycin C inactivated mouse embryonic fibroblasts (MEFs) in DMEM (Gibco, Carlsbad,CA) supplemented with 15% FBS, 10 ng/mL leukemia-inhibiting factor (LIF), penicillin/streptomycin, beta-mercaptoethanol, L-glutamine and non-essential amino-acids. Prior to use in ChIP experiments, V6.5 cells were cultured for two passages under feeder-free conditions. Feeder-independent E14 and EFC-1 mouse embryonic stem cells were grown on 0.1% gelatine. *Ring1A*$^{-/-}$; *RING1B*$^{fl/fl}$; *Rosa26::CreERT2* ESCs were allowed to settle overnight without feeders and cultured in the absence or presence of 4-hydroxy tamoxifen (800 μM) for 48 hr to deplete RING1B. For generation of stable cell lines, 3 μg expression constructs were transfected into E14 or EFC-1 cells using Lipofectamine 2000 (Invitrogen, Carlsbad, CA), and stable integrants were selected using 1.5 μg/mL puromycin.

## Immunofluorescence

Staining was performed in HeLa, 293T, IMR90 and U2OS cells, which were cultured in DMEM (Lonza, Basel, Switzerland) supplemented with 10% FCS and penicillin/streptomycin. Transient transfection in HeLa cells was achieved using Fugene HD reagent (Roche, Basel, Switzerland) and 2 μg of Flag-tagged KDM2B expression plasmid. Briefly, cells seeded on cover slips were fixed with 4% paraformaldehyde for 20 min, permeabilized with 0.5% Triton X-100 for 10 min and blocked using 3% BSA for 30 min. Fixation was done approximately 24 hr after transfections. Cells were incubated with KDM2B rabbit polyclonal primary antibody for 2 hr or the Flag mouse monoclonal M2 antibody (Sigma, St Louis, MO) in case of exogenous expression, followed by incubation in secondary antibody conjugated with Rhodamine or FITC (Jackson ImmunoResearch Laboratories, West Grove, PA) for 1 hr. All intermediary washing steps and dilutions were performed in 1× PBS. After 4,6-diamidino-2-phenylindole dihydrochloride (DAPI) nuclear staining, the cover slips were mounted on glass slides in fluorescent mounting medium (DAKO, Glostrup, Denmark) and allowed to dry overnight. Images were aquired using a Zeiss AxioSkop fluorescent microscope.

## Electrophoretic mobility shift assay (EMSA)

Recombinant ZF-CxxC constructs encompassing human KDM2A (encoding amino acids 558–704) and KDM2B (encoding amino acids 600–750) were engineered to include an N-terminal 6-his tag followed by a tobacco etch virus (TEV) protease cleavage site as previously described (*Blackledge et al., 2010*; *Blackledge et al., 2012*). The ZF-CxxC protein was isolated by Ni-NTA-mediated purification as described before and TEV-cleavage as described previously (*Blackledge et al., 2012*; *Klose and Bird, 2004*). EMSA probes were generated and labelled and EMSA was performed as previously described with samples analysed on a 1.3% agarose gel (*Blackledge et al., 2010*).

## KDM2B knockdown

Scrambled or KDM2B shRNAs were cloned into pLKO.1puro (Addgene, Cambridge, MA) and sequence verified. For producing recombinant lentiviral particles, the shRNA constructs were co-transfected with psPAX2 packaging plasmid and pMD2.G envelope helper plasmid into 293T cells using FuGene (Roche, Basel, Switzerland). The 21mer shRNA sequences used were: (*Kdm2b*) 5′-CGCTGTGGAAATATCTGTCAT-3′, (control) 5′-CCTAAGGTTAAGTCGCCCTCG-3′. Lentiviral infection of E14 cells was performed overnight in the presence of 4 µg/mL polybrene. The following day, cells were diluted into fresh growth media and allowed to settle onto gelatine-coated dishes. Puromycin selection (1.5 µg/mL) was started 48 hr following transduction and stable lines were isolated and expanded.

## KDM2B protein complex purification

Cells were harvested by scraping, washed with 1× PBS and equilibrated in 10 pellet volumes of buffer A (10 mM Hepes pH 7.9, 1.5 mM $MgCl_2$, 10 mM KCl, 0.5 mM DTT, 0.5 mM PMSF, and complete protease inhibitors [Roche, Basel, Switzerland]). The cells were incubated on ice for 10 min and recovered by centrifugation at 1500×*g* for 5 min. The pellet was resuspended in 3 volumes of buffer A supplemented with 0.1% NP-40 and incubated on ice for 10 min with gentle agitation. Nuclei were recovered by centrifugation at 1500× rpm for 5 min. Recovered nuclei were then resuspended in 1× pellet volume of buffer B (5 mM Hepes [pH 7.9], 26% glycerol, 1.5 mM MgCl2, 0.2 mM EDTA, and complete protease inhibitors [Roche], 0.5 mM DTT) supplemented with 400 mM NaCl. The nuclei were first resuspended in a buffer with 250 mM salt. The volume of the buffer and the cells was taken into account, and the salt concentration was increased to 400 mM by gently adding concentrated 5 M NaCl dropwise while mixing. The extraction was allowed to proceed for 1 hr on ice with occasional agitation, and then the nuclei were pelleted at 13,000 rpm for 20 min at 4°C. The supernatant was taken as the nuclear extract.

For purification of KDM2B-Flag/StrepII, the salt concentration of the nuclear extract was first reduced to 150 mM NaCl by dilution into nuclear extraction buffer B without salt. Between 10–15 mg of nuclear extract were used for each large-scale affinity purification. Avidin (IBA, Goettingen, Germany; 20 µg/1 mg extract) was added to remove biotinylated molecules which would bind unspecifically to the matrix. If benzonase treatment was performed, 75 U/mL nuclear extract benzonase nuclease (Novagen, Darmstadt, Germany) was added at this point. Extracts were pre-cleared for 30 min at 4°C, followed by a max speed centrifugation (13000× rpm) for 5 min. Cleared extract was then added to 50 µL packed StrepTactin superflow high-capacity resin (IBA) and allowed to rotate gently for 3 hr at 4°C. Eight wash steps were performed (20 mM Tris pH 8.0, 500 mM NaCl, 0.2% NP-40, 1 mM DTT (fresh) and 5% glycerol) in 1.5 mL low bind tubes (Eppendorf, Hamburg, Germany). Bound material was eluted in buffer containing 20 mM Tris pH 8.0, 150 mM NaCl, 0.2% NP-40, 1 mM DTT (fresh), 5% glycerol and 10 mM D-desthiobiotin (IBA). The same experimental setup was used to identify associated polypeptides for RING1B, YAF2, RYBP and PCGF1.

## Liquid chromatography—tandem mass spectrometry (LC–MS/MS) and data analysis

The immunoprecipitated solution was desalted using chloroform-methanol extraction, followed by overnight in-solution tryptic digestion and subsequent peptide purification using Waters C18 Sep-Pak cartridges. The analysis of immunoprecipitated digested material was performed by LC–MS/MS using an Orbitrap Velos (Thermo mass spectrometer) coupled to a nano-UPLC system (NanoAcquity, Waters, Milford, MA) using a reversed phase 75 µm x 250 mm, 1.7µm particle size column, as described (*Fischer et al., 2012*). MS/MS spectra were searched against the UniProt SwissProt Mus database (v2011.11.18, 16,460 sequences) in Mascot v2.3.01, allowing one missed cleavage and 20 ppm/0.5 Da mass

deviations in MS/MSMS, respectively. Carbamidomethylation of cysteine was a fixed modification. Oxidation of methionine, and deamidation of asparagine and glutamine were used as variable modifications. Protein assignment was based on at least two peptides identified.

## KDM2B immunoprecipitation

For each immunoprecipitation, 0.5 mg nuclear extract was used in buffer BC150 (150 mM KCl, 10% glycerol, 50 mM HEPES [pH 7.9], 0.5 mM EDTA, 0.5 mM DTT [added fresh]). Approximately 3 µg of antibody of interest was added and the reaction allowed to incubate overnight at 4°C. The next morning, 20 µL of packed protein A beads (RepliGen, Waltham, MA) were added, and the samples rotated for 1 hr at 4°C. Following washing with BC300 (300 mM KCl, 10% glycerol, 50 mM HEPES [pH 7.9], 0.5 mM EDTA, 0.5 mM DTT [added fresh]), the beads were resuspended in a small volume of SDS-PAGE loading dye and boiled at 95°C for 5 min. After spinning at 1000×$g$ for 5 min, the supernatant was used for western blotting.

## Gene expression analysis

Total RNA was extracted using the Qiagen RNeasy Mini kit. Approximately 10 µg nucleic acid were treated with Turbo DNase (Ambion, Carlsbad, CA) at 37°C for 30 min, according to the manufacturer's instructions. Genomic DNA-free RNA samples were further purified using the RNeasy kit RNA cleanup protocol. Samples were run on an 1% agarose gel to check quality of RNA preparation and integrity of 18S and 28S rRNA bands. For subsequent RT-PCR analysis, cDNA was synthesized with the ImProm-II Reverse Transcription System (Promega, Madison, WI). Quantitative real-time PCR was performed in duplicate by using Quantace SYBR Green master mix, using *Gapdh* as housekeeping gene.

For microarray studies, RNA integrity was assessed on a BioAnalyzer; all samples had a RNA Integrity Number (RIN) ≥9.5 (Agilent Laboratories, Santa Clara, CA). Sense ssDNA was generated from 300 ng starting RNA with the Ambion WT Expression Kit according to the manufacturer's instructions. Sense ssDNA was fragmented and labeled using the GeneChip WT Terminal Labeling and Controls Kit. The fragmented peak size was measured on the BioAnalyzer and was in the expected 40–70 nt range. The labeled ssDNA from four biological replicates was hybridized to Affymetrix Mouse Gene 1.0 ST Array (Affymetrix, Santa Clara, CA). Chips were processed on an Affymetrix GeneChip Fluidics Station 450 and Scanner 3000. Cel files were generated using Command Console (Affymetrix).

## Microarray analysis

Affymetrix microarray probe intensities were normalised using RMA normalisation via the Bioconductor/R package oligo (version 1.18.1) (*Carvalho and Irizarry, 2010*) and differences tested using the Bioconductor/R package limma (version 3.10.3) (*Smyth, 2004*). In order for expression to have been considered changed, a majority of the probe sets targeting the gene needed a Benjamini and Hochberg corrected FDR of less than 0.05 and the most significant probe required a fold change of at least 1.5-fold between KDM2B knockdown and scrambled control samples. Enrichments in RING1B marked genes were tested by calculating the proportion of genes significantly changed compared to the fraction of genes on the array marked. Differences were tested using the hypergeometric test. Odds ratio for the RING1B marked genes to increase in expression vs decrease was tested using Fisher's exact test.

## Chromatin immunoprecipitation

Chromatin immunoprecipitation was performed as previously described (*Schmidt et al., 2009*), with minor modifications. For KDM2A, KDM2B and RING1B ChIP, cells were fixed for 1 hr in 2 mM EGS, followed by 15 min in 1% formaldehyde, while for histone modification ChIP cells were fixed for 10 min in 1% formaldehyde alone. In both cases, formaldehyde was quenched by the addition of glycine to a final concentration of 125 µM.

Sonication was performed using a BioRuptor sonicator (Diagenode, Liege, Belgium) to produce fragments of approximately 0.5–1 kb. Immunoprecipitation was performed overnight at 4°C with approximately 3 µg of antibody and chromatin corresponding to 5 × 10⁶ cells. Antibody bound proteins were isolated on protein A agarose beads (RepliGen, Waltham, CA) or protein A magnetic Dynabeads (Invitrogen, Carlsbad, CA), washed extensively, eluted, and cross-links reversed according to the Upstate protocol. Samples were then sequentially treated with RNase and proteinase K before being purified on a PureLink

PCR micro column (Invitrogen). Real-time qPCR was performed using Sybr Green (Quantace, London, UK) on a Rotor-Gene 6000 (Corbett, Hilden, Germany) or sequencing libraries were generated as described previously (*Blackledge et al., 2010*) and sequenced on the Illumina HiSeq2000 platform with 51 bp reads.

### ChIP-seq analysis

Published datasets for CpG island (CGI) intervals were obtained from (*Illingworth et al., 2010*). Published RING1B ChIP-seq and input reads from mouse ESCs were obtained from GSM585229 (*Tavares et al., 2012*) and intervals were called as described below. EZH2 ChIP-seq from mouse ESCs was obtained from GSM480161 (*Peng et al., 2009*). TSS annotation and GO slim categories were obtained from Ensembl 66.

All sequence was mapped using Bowtie (*Langmead et al., 2009*) version 0.12.7 against the mouse genome (mm9). Up to two mismatches were allowed and only uniquely mapping reads were kept. Mapped reads were then de-duplicated using Picard to remove potential PCR-duplicates. Where appropriate, sets of mapped reads were normalized by random sampling of the larger set to the size of the smaller. Peak intervals were generated by MACS 1.4.2 (*Zhang et al., 2008*) using normalized chip and input samples. A p-value threshold of $1 \times 10^{-5}$ and a fold enrichment over input threshold of 6 was used to filter the resulting peak intervals. Intervals closer than 200 bp were merged.

Profile plots were generated using the custom script bam2geneprofile by counting reads mapping over each base of a pre-defined set of intervals and averaging the depth. In the case of *Figure 6C*, interval lengths were normalised. In all other cases a fixed window size about a point was used. Where two samples are shown on the same plot, the numbers of mapping reads has been normalised. Results are normalised over the number of intervals in the input set.

To calculate enrichment of KDM2B over KDM2A at CGIs overlapping TSSs, the coverage of each factor which overlapped a 1 kb window around any TSS was calculated using the coverageBed tool from BedTools (*Quinlan and Hall, 2010*) on normalised read sets. For each CGI the ratio of KDM2B to KDM2A reads was calculated. The genes associated with CGIs that showed a greater than twofold enrichment of KDM2B over KDM2A were used to perform GO analysis. Enrichment of GO terms from the Biological Function category of the GO slim hierarchy was tested using the custom script GO.py. The significance of each category was tested using a hypergeometric test and FDRs calculated using the method of Benjamini and Hochberg (*Benjamini and Hochberg, 1995*). Terms with an FDR <0.05 and fold enrichment greater than 3.5 are shown.

ChIP-seq heatmaps were generated using the custom script bam2peakshape.py. Briefly, each CGI window was divided into 25 bp bins and the number of reads mapping to each bin counted. Reads were then normalised to the total number of reads mapping across all intervals. Intervals were then ranked by the number of reads in the RING1B set that map to that interval.

### Source code

Source code for custom scripts described is available in the mercurial repository at www.cgat.org/hg/cgat. The change-set used was 114addb46882.

### Data access

Sequencing and microarray data can be accessed via the geo accession GSE41267.

## Acknowledgements

We would like to thank Ryo Koyama-Nasu for the Human KDM2B cDNA, Jason Carroll for advice on ChIP-sequencing, Dave Brown for comments on the manuscript, Andreas Heger for his work on the CGAT code collection, members of the CGAT group for discussions on data analysis, the Wellcome Trust Integrative Physiology Initiative in Ion Channels and Diseases of Electrically Excitable Cells (OXION) for use of the microarray facility, Dr Nicola Ternette and Dr Roman Fischer for their expert help with the analysis by mass spectrometry, and the Klose and Brockdorff groups for stimulating discussion and provision of reagents. We thank the High-Throughput Genomics Group at the Wellcome Trust Centre for Human Genetics (funded by Wellcome Trust grant reference 090532/Z/09/Z and MRC Hub grantG0900747 91070) for the generation of the Sequencing data.

## Additional information

### Competing interests

CPP: Senior Editor, *eLife*. The remaining authors have declared that no competing interests exist.

### Funding

| Funder | Grant reference number | Author |
|---|---|---|
| Wellcome Trust | WT0834922 | Neil P Blackledge, Hannah K Long, Robert J Klose |
| UK Medical Research Council | | Ian Sudbery, David Sims, Chris P Ponting |
| Cancer Research UK | C28585/A10839 | Anca M Farcas |
| Lister institute of Preventive Medicine | | Anca M Farcas, Neil P Blackledge, Hannah Long, Nathan R Rose, Thomas W Sheahan, Robert J Klose |
| RIKEN | | Haruhiko Koseki |
| Biomedical Research Centre (NIHR), Oxford, UK | | Joanna F McGouran, Benedikt M Kessler |
| Wellcome Trust | WT081385 | Andrea Cerase, Neil Brockdorff |
| St John's College, Oxford | | Nathan R Rose |

The funders had no role in study design, data collection and interpretation, or the decision to submit the work for publication.

### Author contributions

AMF, Conception and design, Acquisition of data, Analysis and interpretation of data, Drafting or revising the article; NPB, Conception and design, Acquisition of data, Analysis and interpretation of data, Drafting or revising the article; IS, Conception and design, Analysis and interpretation of data, Drafting or revising the article; HKL, Conception and design, Acquisition of data, Analysis and interpretation of data, Drafting or revising the article; JFM, Acquisition of data, Analysis and interpretation of data; NRR, Conception and design, Acquisition of data, Analysis and interpretation of data; SL, Conception and design, Acquisition of data, Analysis and interpretation of data; DS, Conception and design, Analysis and interpretation of data, Drafting or revising the article; AC, Acquisition of data, Contributed unpublished essential data or reagents; TWS, Acquisition of data, Contributed unpublished essential data or reagents; HK, Drafting or revising the article, Contributed unpublished essential data or reagents; NB, Drafting or revising the article, Contributed unpublished essential data or reagents; CPP, Conception and design, Analysis and interpretation of data, Drafting or revising the article; BMK, Conception and design, Acquisition of data, Analysis and interpretation of data, Drafting or revising the article; RJK, Conception and design, Acquisition of data, Analysis and interpretation of data, Drafting or revising the article

## Additional files

### Major datasets

The following datasets were generated

| Author(s) | Year | Dataset title | Dataset ID and/or URL | Database, license, and accessibility information |
|---|---|---|---|---|
| Farcas AM, Blackledge NP, Sudbery I, Long HK, McGouran JF, Rose NR, Lee S, Sims D, Cerase A, Sheahan TW, Koseki H, Brockdorff N, Ponting CP, Kessler BM, Klose RJ | 2012 | KDM2B binds CpG islands and modulates recruitment of Ring1b | GSE41267, http://www.ncbi.nlm.nih.gov/geo/query/acc.cgi?acc=GSE41267 | In the public domain at GEO: http://www.ncbi.nlm.nih.gov/geo/. |

**Reporting Standards:** N/A

The following previously published datasets were used:

| Author(s) | Year | Dataset title | Dataset ID and/or URL | Database, license, and accessibility information |
|---|---|---|---|---|
| Illingworth RS, Gruenewald-Schneider U, Webb S, Kerr A, James KD, Turner DJ, Smith C, Harrison DJ, Andrews R, Bird AP | 2010 | Comprehensive analysis of orphan CpG islands identifes novel promoters with conserved DNA methylation dynamics | Series GSE21442 http://www.ncbi.nlm.nih.gov/geo/query/acc.cgi?acc=GSE21442 | In the public domain at GEO: http://www.ncbi.nlm.nih.gov/geo/. |
| **Reporting Standards:** N/A | | | | |
| Tavares L, Dimitrova E, Oxley D, Webster J, Poot R, Demmers J, Bezstarosti K, Taylor S, Ura H, Koide H, Wutz A, Vidal M, Elderkin S, Brockdorff N | 2011 | H3K27me3 is not required for recruitment of Polycomb repressor complex 1 to target loci in mouse embryonic stem cells. (Sample: EedWT-Ring1B) | Series GSE23716 Sample GSM585229 http://www.ncbi.nlm.nih.gov/geo/query/acc.cgi?acc=gsm585229 | In the public domain at GEO: http://www.ncbi.nlm.nih.gov/geo/. |
| **Reporting Standards:** N/A | | | | |
| Peng JC, Valouev A, Swigut T, Zhang J, Zhao Y, Sidow A, Wysocka J | 2009 | Jarid2/Jumonji coordinates control of PRC2 enzymatic activity and target gene occupancy in pluripotent cells. (Sample: Ezh2_ChIPSeq) | Series GSE18776 Sample GSM480161 http://www.ncbi.nlm.nih.gov/geo/query/acc.cgi?acc=gsm480161 | In the public domain at GEO: http://www.ncbi.nlm.nih.gov/geo/. |
| **Reporting Standards:** N/A | | | | |

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
