## [Decision Letter]

Thank you for choosing to send your work entitled "KDM2B links the Polycomb Repressive Complex 1 (PRC1) to recognition of CpG islands" for consideration at *eLife*. Your article has been evaluated by a Senior Editor and 3 reviewers, one of whom, Kevin Struhl, is a member of our Board of Reviewing Editors.

The Reviewing Editor and the other reviewers discussed their comments before we reached this decision, and the Reviewing Editor has assembled the following comments based on the reviewers' reports.

Repression by polycomb complexes is of high biological importance yet is not well understood mechanistically. An important question, particularly in mammalian cells, is how Pc complexes are recruited to specific sites. Here, the authors provide evidence that the histone demethylase KDM2B recruits the PRC1 complex to CpG islands. The reviewers agree that the work is interesting and the manuscript is acceptable in principle for publication in *eLife*. No new experiments are needed. However, it is essential to address the following comments in the text:

* What are H3K36 demethylases doing at sites not expected to contain H3K36 methylation?

* Is this recruitment mechanism typical or is it specific for ESC cells?

* What excludes KDM2A from Polycomb target CGIs?

* What PRC1 protein(s) does KDM2B interact with? With what protein domain? Why does it not recruit CBX-containing complexes?

* Only a small subset of KDM2B binding sites are also occupied by RING1B so other features or factors must be involved in the recruitment of PRC1. Please discuss them.

* How is PRC2 recruited to KDM2B/RING1B sites? It seems to be there according to Figure 2, for example. Is a CBX protein associated with these sites? Is there H3K27me3 associated with KDM2B/RING1B binding sites?

* It is unclear why, if KDM2B is required for all Ring1b binding sites, loss of KDM2B only derepresses a very small subset. If Ring1b is found at 70% of all genomic CpG islands, it must be found also at sites that bind KDM2A. Does RING1B antagonize binding of KDM2A? Does KDM2A now bind to these CpG islands when no PRC1 is present?

* Without a mechanism to recruit PRC2 to the same sites, the model is not very explanatory. Furthermore, nothing was said about recruitment of CBX-containing PRC1. Evidently these are not recruited by KDM2B. So there are several links missing in the causal chain in the scenario proposed, which should be accounted for.

* How are Polycomb-repressed target genes identified? By the binding of Ring1b? By H3K27me3?

* The loss of H2Aub even from those genes that are supposed to be derepressed after KDM2B knockdown is not impressive: in most cases it is no more than 20% lower. That means that the 40% loss in global H2Aub is not concentrated in these genes and suggests that KDM2B plays a minor role in recruiting some kind of PRC1. Things are even worse at PcG target CGIs that do not become derepressed.

* There continues to be a misguided insistence on the assumption that binding PRC1 automatically compacts chromatin, although in ES cells it is clear that many binding sites are the typical bivalent sites that also have methylated H3K4 and bind RNA polymerase.

---

## [Author Response]

In responding to the reviewers’ comments about some of the overlaps between CpG islands (CGIs) and PRC1/2 intervals we noticed that the published CGI dataset (Illingworth et al 2010) used throughout our study had a small number of CGI intervals (172 out 23021) that overlapped. Therefore these overlapping intervals should have been considered as one individual interval, effectively reducing the total number of CGIs in the dataset from 23021 to 22849. For the sake of accuracy we have merged these overlapping CGI intervals in the resubmission. This does not affect our interpretation or observations, but this did necessitate redrawing Figure 2A and B. The changes in Figure 2A are not visible by eye but we wanted to make you aware that this figure had been redrawn since the original submission.

** What are H3K36 demethylases doing at sites not expected to contain H3K36 methylation*?

KDM2A and B are histone demethylases that remove the H3K36me1/me2 modifications. In mammals our ChIP analysis (Blackledge NP et al 2010) indicates that the H3K36me1/me2 modifications cover most of the genome and global mass spectrometry suggests H3K36me1/me2 is found on up to ∼60% of all histone H3 (Peters AH et al 2003 and Robin P et al 2007). Interestingly, there is a specific reduction in H3K36me1/me2 at CpG island regions and we recently showed that demethylation by KDM2A contributes to this depletion (Blackledge NP et al 2010). This suggests that a certain amount of H3K36me1/me2 is placed at CGIs but is then actively removed by targeting of demethylase activity. This may function as a way of protecting these CGI associated gene promoter regions from the inhibitory effects of H3K36me1/me2, which is proposed to counteract transcription (Strahl BD et al 2002 and Carrozza MJ et all 2005). We have now included a section in the introduction to clarify this point as follows:

‘In mammals H3K36me1/me2 is found broadly across the genome […] and may be inhibitory to transcriptional initiation […]. Specific targeting of KDM2A to CGIs appears to remove H3K36me2 at CGIs as part of a mechanism to mark these regions with transcriptionally permissive chromatin.’

** Is this recruitment mechanism typical or is it specific for ESC cells*?

We think that this recruitment mechanism will also function in other cell types as KDM2B has been identified in purifications of PRC1 from human and mouse cancer cell lines. However, we have not formally tested this possibility. Therefore, we have added a section to the discussion that brings attention to this point:

‘In order to fully realize the complexities and interplay between polycomb silencing activities, more detailed molecular and biochemical understanding of this system is clearly required. For example, our study has focussed on ESCs where polycomb factors are highly abundant, so it will be imperative to understand if similar KDM2B dependent PRC1 targeting mechanisms function in more committed cell types.’

** What excludes KDM2A from Polycomb target CGIs*?

Our recent work demonstrated that KDM2A requires accessible linker DNA to associate with chromatin in vitro and in vivo (Zhou et al 2012). Based on these observations, one possibility may be that linker DNA is more occluded at CGIs with high levels of polycomb occupancy. We discuss this in the results section titled ‘KDM2B relies on its ZF-CxxC domain and not its interaction with PRC1 to bind polycomb repressed CGIs’. In future studies we aim to understand in detail the molecular nature of this exclusion.

** What PRC1 protein(s) does KDM2B interact with? With what protein domain? Why does it not recruit CBX-containing complexes*?

Future work is planned to map in detail the protein interactions within the KDM2B-PRC1 variant complex, but this is outside the scope of the current manuscript. Our work in mouse ES cells and recently published work from the Reinberg lab using human cancer cell lines indicate that individual PCGF factors may dictate the capacity of the RING1B subunit in PRC1 complexes to associate with either the RYBP/YAF2 or CBX factors. The molecular/structural explanation for these differences remains unknown. To highlight the lack of CBX proteins in the KDM2B variant PRC1 complex in embryonic stem cells and to draw attention to the potential role of PCGF1 in specifying this, we have added the following statement to the results section describing Figure 3:

‘Interestingly, in agreement with PRC1 purifications from cancer cell lines […], the PCGF1-containing complex associates with RYBP or YAF2 but fails to integrate chromobox domain-containing (CBX) proteins that recognize H3K27me3 […]. CBX proteins and RYBP/YAF2 are thought to interact with RING1B in a mutually exclusive manner. This suggests that PCGF1 may play an important role in specifying the capacity of RYBP and YAF2 to associate with RING1B in the KDM2B variant PRC1 complex.’

** Only a small subset of KDM2B binding sites are also occupied by RING1B so other features or factors must be involved in the recruitment of PRC1. Please discuss them*.

Some CGIs appear to have high levels of PRC1 occupancy (‘established’ polycomb targets). Interestingly, we discover that there is also low but appreciable levels of RING1B at over 70% of CGIs, and we refer to these as ‘novel’ low magnitude PRC1 occupied sites. Therefore most KDM2B bound CGIs also show PRC1 occupancy (i.e., not a small subset). This is in fitting with the physical association between KDM2B and the PRC1 variant complex. As suggested by the reviewers, there are other mechanisms that contribute to PRC1 targeting and we cover these extensively in the discussion including; (1) The hierarchal recruitment mechanism based on CBX protein dependent recognition of H3K27me3, (2) The potential involvement of RYBP through poorly defined molecular mechanisms, and (3) the direct targeting of PRC1 by transcription factors like REST and GATA1.

** How is PRC2 recruited to KDM2B/RING1B sites? It seems to be there according to Figure 2, for example. Is a CBX protein associated with these sites? Is there H3K27me3 associated with KDM2B/RING1B binding sites*?

It is still unclear how PRC2 is recruited to polycomb targets and CGIs, and we don’t attempt to address this question in this study. We have now analysed a published CBX7 ChIP-seq data in ES cells and observe very little overlap of CBX7 with novel RING1B occupied CGIs. Although this suggests that CBX7 is largely absent, we have not included this analysis, as the signal to noise in these published ChIP-seq experiments is not sufficient in our opinion to accurately identify intervals. We have however analysed the overlap with EZH2 with novel RING1B CGIs (which are mostly occupied by KDM2B) and observe that only 23% of these novel sites are occupied by detectable levels of EZH2 and 20% with H3K27me3. We have now stated this in the results section as follows:

‘Importantly, a minority of these novel RING1B occupied CGI sites showed detectable EZH2 (23%) or H3K27me3 (20%), indicating that these sites are largely devoid of appreciable PRC2.’

** It is unclear why, if KDM2B is required for all Ring1b binding sites, loss of KDM2B only derepresses a very small subset*.

This is a very interesting point. One possibility that would explain the reactivation of only a small number of polycomb target genes is that the shRNA based knockdown of KDM2B results in a hypomorphic situation. This may lead to a reactivation of a subset of genes that are most sensitive to PRC1 levels. Another possibility is that silencing at ‘established’ polycomb repressed genes is only partially sensitive to the loss of KDM2B due to the compensatory effects of other PRC1 complexes. In future work, null alleles for KDM2B will aid in answering these questions more definitively. Nevertheless we have added a section to the discussion to address this point more directly:

‘[…] knockdown of KDM2B leads to reactivation of only a subset of polycomb repressed genes. Although this may be due to incomplete removal of KDM2B by RNAi, it may also be related to compensatory function of other PRC1 complexes.‘

*If Ring1b is found at 70% of all genomic CpG islands, it must be found also at sites that bind KDM2A. Does RING1B antagonize binding of KDM2A? Does KDM2A now bind to these CpG islands when no PRC1 is present*?

ChIP-seq analysis indicates that enrichment of KDM2A, KDM2B, and PRC1 can be detected at many CGIs. We are unsure if RING1B directly antagonizes KDM2A binding to CGIs. We agree this is an interesting question and future work is planned to examine in more detail the relationship between KDM2A and PRC1 (RING1B), particularly at the molecular and biochemical level.

** Without a mechanism to recruit PRC2 to the same sites, the model is not very explanatory. Furthermore, nothing was said about recruitment of CBX-containing PRC1. Evidently these are not recruited by KDM2B. So there are several links missing in the causal chain in the scenario proposed, which should be accounted for*.

We agree that KDM2B dependent targeting of PRC1 is on its own not sufficient to explain the full functionality of PRC1 complexes in vivo and the over all activity of PRC1/2 at ‘established’ polycomb repressed sites as indicated by the reviewers. We attempted to make this clear in the initial submission by discussing the CBX dependent hierarchical mode of targeting which highlights the fact that other mechanisms exist to target PRC1. To make this point more clear in the resubmission, and specifically draw attention to the fact that KDM2B is insufficient to account for the ‘causal chain’ of events required for PRC1 dependent activity in vivo, we have now added a section to the discussion covering this important point more succinctly:

‘Also, it is known that H3K27me3 and CBX dependent PRC1 targeting activities play an important role in RING1B occupancy. Therefore, KDM2B dependent targeting of the PCGF1-PRC1 complex on its own is insufficient to account for the functionality of PRC1 in vivo. This suggests that a complex set of PRC1 targeting mechanisms, perhaps utilizing specific PRC1 complexes, and based on both direct and indirect recognition of CGI associated sites is required for normal PRC1 function.’

** How are Polycomb-repressed target genes identified? By the binding of Ring1b? By H3K27me3*?

This is a very important question and is not resolved by our study. Therefore, we speculate in our discussion that KDM2B may contribute a sampling mechanism that brings PRC1 to CGIs and the local balance of activators/repressors may define whether polycomb can ‘establish’ silencing. This general principle is in agreement with the suggested function of PREs in *Drosophila*, which are regulated by a balance between trithorax and polycomb activities.

** The loss of H2Aub even from those genes that are supposed to be derepressed after KDM2B knockdown is not impressive: in most cases it is no more than 20% lower. That means that the 40% loss in global H2Aub is not concentrated in these genes and suggests that KDM2B plays a minor role in recruiting some kind of PRC1. Things are even worse at PcG target CGIs that do not become derepressed*.

We agree that the effects of KDM2B depletion on uH2A levels at ‘established’ polycomb repressed CGIs are modest and that the largest changes occur at reactivated genes. As the reviewers point out, this is interesting as the global levels of uH2A reduction seem to be much more dramatic (40%). We speculate that this may be related to uH2A being placed at low but significant levels at PRC1 associated CGIs that don’t have ‘established’ polycomb repression (i.e., ‘novel’ RING1B sites). We have now added a statement to clarify this in the results section:

‘Interestingly, the level of uH2A reduction at individual loci did not appear as dramatic as the global reductions observed by western blotting. One possibility for this difference may be that pervasive KDM2B dependent targeting of PRC1 to CGIs leads to low level deposition of uH2A away from established polycomb sites. One can envisage how loss of KDM2B would more dramatically affect this additional pool of uH2A resulting in significant global reduction in its abundance.’

** There continues to be a misguided insistence on the assumption that binding PRC1 automatically compacts chromatin, although in ES cells it is clear that many binding sites are the typical bivalent sites that also have methylated H3K4 and bind RNA polymerase*.

We use the term compact(ed) three times in the manuscript. In the first two instances it is to describe the published observations that PRC1 occupied chromatin is compact. This is supported by in vivo work using DNA based fluorescent in site hybridization (Eskeland R et al 2010) and in vitro observations that PRC1 compacts chromatin (Francis NJ et al 2004). The third instance, which is found in the discussion, has now been reworded. Previously we said in discussing a speculative model for polycomb activity that, PRC2 might favour ‘the increasingly compacted chromatin state *created by* PRC1 occupancy’. We have now revised this to read: ‘favouring the increasingly compacted chromatin state *associated with* PRC1 occupancy’. We believe it is important and reasonable to discuss the concept that chromatin compaction may be related to PRC1 function at CGIs, but agree with the reviewers that the published literature indicates that a compacted state is ‘associated with’ as opposed to ‘created by’ PRC1 occupancy.